Subject Areas:
complexity/applied mathematics

Keywords:
networks, geometry, Ricci curvature, financial network, stock market

Authors for correspondence:
Areejit Samal
e-mail: asamal@imsc.res.in
Anirban Chakraborti
e-mail: anirban@jnu.ac.in

†These authors contributed equally to this study.

# Network geometry and market instability

Areejit Samal[1,2,†], Hirdesh K. Pharasi[3,†], Sarath Jyotsna Ramaia[4], Harish Kannan[5], Emil Saucan[6], Jürgen Jost[7,8] and Anirban Chakraborti[9,10,11]

[1]The Institute of Mathematical Sciences (IMSc), Chennai 600113, India
[2]Homi Bhabha National Institute (HBNI), Mumbai 400094, India
[3]Instituto de Ciencias Físicas, Universidad Nacional Autónoma de México, Cuernavaca 62210, Mexico
[4]Department of Applied Mathematics and Computational Sciences, PSG College of Technology, Coimbatore 641004, India
[5]Department of Mathematics, University of California San Diego, La Jolla, California 92093, USA
[6]Department of Applied Mathematics, ORT Braude College, Karmiel 2161002, Israel
[7]Max Planck Institute for Mathematics in the Sciences, Leipzig 04103, Germany
[8]The Santa Fe Institute, Santa Fe, NM 87501, USA
[9]School of Computational and Integrative Sciences, Jawaharlal Nehru University, New Delhi 110067, India
[10]Centre for Complexity Economics, Applied Spirituality and Public Policy (CEASP), Jindal School of Government and Public Policy, O.P. Jindal Global University, Sonipat 131001, India
[11]Centro Internacional de Ciencias, Cuernavaca 62210, Mexico

 AS, 0000-0002-6796-9604; JJ, 0000-0001-5258-6590; AC, 0000-0002-6235-0204

The complexity of financial markets arise from the strategic interactions among agents trading stocks, which manifest in the form of vibrant correlation patterns among stock prices. Over the past few decades, complex financial markets have often been represented as networks whose interacting pairs of nodes are stocks, connected by edges that signify the correlation strengths. However, we often have interactions that occur in groups of three or more nodes, and these cannot be described simply by pairwise interactions but we also need to take the relations between these interactions into account. Only recently, researchers have started devoting attention to the higher-order architecture of complex financial systems, that can significantly enhance our ability to estimate systemic risk as well as measure the robustness of financial systems in terms of market efficiency. Geometry-inspired network measures, such as the Ollivier–Ricci curvature and Forman–Ricci curvature, can be used to capture the network fragility and continuously monitor financial dynamics. Here, we explore the utility of such discrete Ricci curvatures in characterizing the structure of financial systems, and further, evaluate them as generic indicators of the market instability. For this purpose, we examine the daily returns from a set of stocks comprising the

USA S&P-500 and the Japanese Nikkei-225 over a 32-year period, and monitor the changes in the edge-centric network curvatures. We find that the different geometric measures capture well the system-level features of the market and hence we can distinguish between the normal or 'business-as-usual' periods and all the major market crashes. This can be very useful in strategic designing of financial systems and regulating the markets in order to tackle financial instabilities.

## 1. Introduction

For centuries, science had thrived on the method of reductionism—considering the units of a system in isolation, and then trying to understand and infer about the whole system. However, the simple method of reductionism has severe limitations [1], and fails to a large extent when it comes to the understanding and modelling the collective behaviour of the components of a 'complex system'. More and more systems are now being identified as complex systems, and hence scientists are now embracing the idea of complexity as one of the governing principles of the world we live in. Any deep understanding of a complex system has to be based on a system-level description, since a key ingredient of any complex system is the rich interplay of nonlinear interactions between the system components. The financial market is truly a spectacular example of such a complex system, where the agents interact strategically to determine the best prices of the assets. So new tools and interdisciplinary approaches are needed [2,3], and already there has been an influx of ideas from econophysics and complexity theory [4–8] to explain and understand economic and financial markets.

The traditional economic theories, based on axiomatic approaches and consequently less predictive power, could not foresee an event like the sub-prime crisis of 2007–2008 or the long-lasting effects of such a critical financial crash on the global economy. Researchers advocated that new concepts and techniques [9] like tipping points, feedback, contagion, network analysis along with the use of complexity models [10] could help in better understanding of highly interconnected economic and financial systems, as well as monitoring them. There have been numerous papers in the past that have addressed similar concerns and tried to adopt new approaches for studying financial systems. Since the correlations among stocks change with time, the underlying market dynamics generate very interesting correlation-based networks evolving over time. The study of empirical cross-correlations among stock prices goes back to more than two decades [11–16]. One of commonly adopted approaches for the modelling and analysis of complex financial systems has been correlation-based networks, and it has emerged as an important tool [11,12,17–22].

A network or graph consists of nodes connected by edges. In real-world networks, nodes represent the components or entities, while edges represent the interactions or relationships between nodes. In the context of financial markets, the nodes represent the stocks and the edges characterize the correlation strengths (or their transformations into distance measures). The network formed by connecting stocks of highly correlated prices, price returns and trading volumes are all scale-free, with a relatively small number of stocks influencing the majority of the stocks in the market [23]. Hierarchical clustering has been used to cluster stocks into sectors and sub-sectors, and their network analysis provides additional information on the interrelationships among the stocks [24,25]. The cross-correlations among stock returns allow one to construct other correlation-based networks such as minimum spanning tree (MST) [11,12,18,26] or a threshold network [27]. Another approach to monitor the correlation-based networks over time, referred to as structural entropy, quantifies the structural changes of the network as a single parameter. It takes into account the number of communities as well as the size of the communities [28] to determine the structural entropy, which is then used to continuously monitor the market. The thermodynamical entropy [29] can also be used to describe the dynamics of stock market networks as it acts like an indicator for the financial system. Very recently, based on the distribution properties of the eigenvector centralities of correlation matrices, Chakraborti & Pharasi [30] have proposed a computationally cheap yet uniquely defined and non-arbitrary eigen-entropy measure, to show that the financial market undergoes 'phase separation' and there exists a new type of scaling behaviour (data collapse) in financial markets. Further, a recent review by Kukreti et al. [31] critically examines correlation-based networks and entropy approaches in evolving financial systems. To understand the topology of the correlation-based networks as well as to define the complexity, a volume-based dimension has also been proposed by Nie et al. [32]. There have also been some novel studies where the financial market has been considered as a quasi-stationary system, and then the ensuing dynamics have been studied [33–37].

Introduced long ago by Gauss & Riemann, curvature is a central concept in geometry that quantifies the extent to which a space is curved [38]. In geometry, the primary invariant is curvature in its many forms. While curvature has connections to several essential aspects of the underlying space, in a specific case, curvature has a connection to the Laplacian, and hence, to the 'heat kernel' on a network. Curvature also has connections to the Brownian motion and entropy growth on a network. Moreover, curvature is also related to algebraic topological aspects, such as the homology groups and Betti numbers, which are relevant, for instance, for persistent homology and topological data analysis [39]. Recently, there has been immense interest in geometrical characterization of complex networks [40–44]. Network geometry can reveal higher-order correlations between nodes beyond pairwise relationships captured by edges connecting two nodes in a graph [45–47]. From the point of view of structure and dynamics of complex networks, edges are more important than nodes, since the nodes by themselves cannot constitute a meaningful network. Hence, it may be more important to develop edge-centric measures rather than node-centric measures to characterize the structure of complex networks [43,48].

Surprisingly, geometrical concepts, especially, discrete notions of Ricci curvature, have only very recently been used as edge-centric network measures [42,43,48–51]. Furthermore, curvature has deep connections to related evolution equations that can be used to predict the long-time evolution of networks. Although the importance of geometric measures like curvature have been understood for quite some time, yet there has been limited number of applications in the context of complex financial networks. In particular, Sandhu *et al*. [50] studied the evolution of Ollivier–Ricci curvature [52,53] in threshold networks for the USA S&P-500 market over a 15-year span (1998–2013) and showed that the Ollivier–Ricci curvature is correlated to the increase in market network fragility. Consequently, Sandhu *et al*. [50] suggested that the Ollivier–Ricci curvature can be employed as an indicator of market fragility and study the designing of (banking) systems and framing regulation policies to combat financial instabilities such as the sub-prime crisis of 2007–2008. In this paper, we expand the study of geometry-inspired network measures for characterizing the structure of the financial systems to four notions of discrete Ricci curvature, and evaluate the curvature measures as generic indicators of the market instability.

It is noteworthy that in the present paper, the term 'curvature' refers to four notions of discrete Ricci curvature investigated here, which are as such intrinsic curvatures, and not extrinsic curvatures as has been considered elsewhere in the context of complex networks (e.g. [54]). Recall that extrinsic geometry is given by embedding the networks in a suitable ambient space (which in practice is the hyperbolic plane or space), and thereafter, the geometric properties induced by the embedding space are studied (e.g. [55]). While this approach is intuitive and conducive to simple illustrations, such network embeddings are distorting, except for the special case of isometric embeddings. By contrast, the intrinsic approach to networks is independent of any specific embedding, and hence, of the necessary additional computations and any distortion. Moreover, such an intrinsic approach allows for the independent study of such powerful tools as the Ricci flow, without the vagaries associated with the embedding in an ambient space of certain dimension (e.g. [56]). Furthermore, the Ollivier–Ricci curvature has been employed to show that the 'backbone' of certain real-world networks is indeed tree-like, hence intrinsically hyperbolic [49]. Specific to financial networks, Sandhu *et al*. [50] have shown that Ollivier–Ricci curvature, which is of course an intrinsic curvature, presents a powerful tool in the detection of financial market crashes. In this work, we have considered three additional notions [43,55] of discrete Ricci curvature for the study of financial networks.

In the present paper, we examine the daily returns from a set of stocks comprising the USA S&P-500 and the Japanese Nikkei-225 over a 32-year period, and monitor the changes in the edge-centric geometric curvatures. A major goal of this research is to evaluate different notions of discrete Ricci curvature for their ability to unravel the structure of complex financial networks and serve as indicators of market instabilities. Our study confirms that during a normal period the market is very modular and heterogeneous, whereas during an instability (crisis) the market is more homogeneous, highly connected and less modular [18,21,22,57]. Further, we find that the discrete Ricci curvature measures, especially Forman–Ricci curvature [43,48], capture well the system-level features of the market and hence we can distinguish between the normal or 'business-as-usual' periods and all the major market crises (bubbles and crashes). Importantly, among four Ricci-type curvature measures, the Forman–Ricci curvature of edges correlates highest with the traditional market indicators and acts as an excellent indicator for the system-level fear (volatility) and fragility (risk) for both the markets. We also find using these geometric measures that there are succinct and inherent differences in the two markets, USA S&P-500 and Japan Nikkei-225. These new insights will help us to understand

tipping points, systemic risk, and resilience in financial networks, and enable us to develop monitoring tools required for the highly interconnected financial systems and perhaps forecast future financial crises and market slowdowns.

# 2. Ricci-type curvatures for edge-centric analysis of networks

The classical notion of Ricci curvature applies to smooth manifolds, and its classical definition requires tensors and higher-order derivatives [38]. Thus, the classical definition of Ricci curvature is not immediately applicable in the discrete context of graphs or networks. Therefore, in order to develop any meaningful notion of Ricci curvature for networks, one has to inspect the essential geometric properties captured by this curvature notion, and find their proper analogues for discrete networks. To this end, it is essential to recall that Ricci curvature quantifies two essential geometric properties of the manifold, namely, volume growth and dispersion of geodesics. See electronic supplementary material, figure S1 for a schematic illustration of the Ricci curvature. Further, since classical Ricci curvature is associated with a vector (direction) in smooth manifolds [38], in the discrete case of networks, it is naturally assigned to edges [48]. Thus, notions of discrete Ricci curvatures are associated with edges rather than vertices or nodes in networks [48]. Note that no discretization of Ricci curvature for networks can capture the full spectrum of properties of the classical Ricci curvature defined on smooth manifolds, and thus, each discretization can shed a different light on the analysed networks [48]. In this work, we apply four notions of discrete Ricci curvature for networks to study the correlation-based networks of stock markets.

## 2.1. Ollivier–Ricci curvature

Ollivier's discretization [52,53] of the classical Ricci curvature has been extensively used to analyse graphs or networks [42,48–51,58–62]. Ollivier's definition is based on the following observation. In spaces of positive curvature, balls are closer to each other on the average than their centres, while in spaces of negative curvature, balls are farther away on the average than their centres (electronic supplementary material, figure S2). Ollivier's definition extends this observation from balls (volumes) to measures (probabilities). More precisely, the Ollivier–Ricci (OR) curvature of an edge $e$ between nodes $u$ and $v$ is defined as

$$\mathbf{O}(e) = 1 - \frac{W_1(m_u, m_v)}{d(u, v)}, \tag{2.1}$$

where $m_u$ and $m_v$ represent measures concentrated at nodes $u$ and $v$, respectively, $W_1$ denotes the Wasserstein distance [63] (also known as the earth mover's distance) between the discrete probability measures $m_u$ and $m_v$, and the cost $d(u, v)$ is the distance between nodes $u$ and $v$, respectively. Moreover, the Wasserstein distance $W_1(m_u, m_v)$, which gives the transportation distance between the two measures $m_u$ and $m_v$, is given by

$$W_1(m_u, m_v) = \inf_{\mu_{u,v} \in \prod(m_u, m_v)} \sum_{(u', v') \in V \times V} d(u', v') \mu_{u,v}(u', v'), \tag{2.2}$$

with $\prod(m_u, m_v)$ being the set of probability measures $\mu_{u,v}$ that satisfy

$$\sum_{v' \in V} \mu_{u,v}(u', v') = m_u(u'), \quad \sum_{u' \in V} \mu_{u,v}(u', v') = m_v(v') \tag{2.3}$$

where $V$ is the set of nodes in the graph. The above equation represents all the transportation possibilities of the mass $m_u$ to $m_v$. $W_1(m_u, m_v)$ is the minimal cost or distance to transport the mass of $m_u$ to that of $m_v$. Note that the distance $d(u', v')$ in equation (2.2) is taken to be the path distance in the unweighted or weighted graph. Furthermore, the probability distribution $m_u$ for $u \in V$ has to be specified, and this is chosen to be uniform over neighbouring nodes of $u$ [59].

Simply stated, to determine the OR curvature of an edge $e$, in equation (2.1) one compares the average distance between the neighbours of the nodes $u$ and $v$ anchoring the edge $e$ in an optimal arrangement with the distance between $u$ and $v$ itself. Importantly, the average distance between neighbours of $u$ and $v$ is evaluated as an optimal transport problem wherein the neighbours of $u$ are coupled with those of $v$ in such a manner that the average distance is as small as possible. In the setting of discrete graphs or networks, OR curvature by definition captures the volume growth aspect of the classical notion for

smooth manifolds (e.g. [48] for details). In this work, we have computed the average OR curvature of edges (ORE) in undirected and weighted networks using equation (2.1).

## 2.2. Forman–Ricci curvature

Forman's approach to the discretization of Ricci curvature [64] is more algebraic in nature and is based on the relation between the Riemannian Laplace operator and Ricci curvature. While devised originally for a much larger class of discrete geometric objects than graphs, an adaptation to network setting was recently introduced by some of us [43]. The Forman–Ricci (FR) curvature $\mathbf{F}(e)$ of an edge $e$ in an undirected network with weights assigned to both edges and nodes is given by [43]

$$\mathbf{F}(e) = w_e \left( \frac{w_{v_1}}{w_e} + \frac{w_{v_2}}{w_e} - \sum_{e_{v_1} \sim e,\, e_{v_2} \sim e} \left[ \frac{w_{v_1}}{\sqrt{w_e w_{e_{v_1}}}} + \frac{w_{v_2}}{\sqrt{w_e w_{e_{v_2}}}} \right] \right), \qquad (2.4)$$

where $e$ denotes the edge under consideration between nodes $v_1$ and $v_2$, $w_e$ denotes the weight of the edge $e$, $w_{v_1}$ and $w_{v_2}$ denote the weights associated with the nodes $v_1$ and $v_2$, respectively, $e_{v_1} \sim e$ and $e_{v_2} \sim e$ denote the set of edges incident on nodes $v_1$ and $v_2$, respectively, after excluding the edge $e$ under consideration which connects the two nodes $v_1$ and $v_2$. Furthermore, some of us have also extended the notion of FR curvature to directed networks [65]. In the case of discrete networks, FR curvature captures the geodesic dispersal property of the classical notion [48]. In electronic supplementary material, figure S3, we illustrate, using a simple example, the computation of FR curvature in an undirected graph. In this work, we have computed the average FR curvature of edges (FRE) in undirected and weighted networks using equation (2.4).

From a geometric perspective, the FR curvature quantifies the information spread at the ends of edges in a network (figure 1; electronic supplementary material, figure S3). The higher the information spread at the ends of an edge, the more negative will be the value of its FR curvature. Specifically, an edge with high negative FR curvature is likely to have several neighbouring edges connected to both anchoring nodes, and moreover, such an edge can be seen as a funnel at both ends, connecting many other nodes. Intuitively, such an edge with high negative FR curvature can be expected to have high edge betweenness centrality, as many shortest paths between other nodes, including those quite far in the network, are also likely to pass through this edge. Previously, some of us have empirically shown a high statistical correlation between FR curvature and edge betweenness centrality in diverse networks [48,66].

## 2.3. Menger–Ricci curvature

The remaining two curvatures studied here are adaptations of curvatures for metric spaces to discrete graphs. Indeed, both unweighted and weighted graphs can be viewed as a metric space where the distance between any two nodes can be specified by the path length between them. Among notions of metric, and indeed, discrete curvature, Menger [67] has proposed the simplest and earliest definition whereby he defines the curvature of metric triangles $T$ formed by three points in the space as the reciprocal $1/R(T)$ of the radius $R(T)$ of the circumscribed circle of a triangle $T$. Recently, some of us [55,68] have adapted Menger's definition to networks. Let $(M, d)$ be a metric space and $T = T(a, b, c)$ be a triangle with sides $a$, $b$, $c$, then the Menger curvature of $T$ is given by

$$K_M(T) = \frac{\sqrt{p(p-a)(p-b)(p-c)}}{a \cdot b \cdot c}, \qquad (2.5)$$

where $p = (a + b + c)/2$. In the particular case of a combinatorial triangle with each side of length 1, the above formula gives $K_M(T) = \sqrt{3}/2$. Furthermore, it is clear from the above formula that Menger curvature is always positive. Following the differential geometric approach, the Menger–Ricci (MR) curvature of an edge $e$ in a network can be defined as [55,68]

$$\kappa_M(e) = \sum_{T_e \sim e} \kappa_M(T_e), \qquad (2.6)$$

where $T_e \sim e$ denote the triangles adjacent to the edge $e$. Intuitively, if an edge is part of several triangles in the network, such an edge will have high positive MR curvature (figure 1). In electronic supplementary material, figure S4, we illustrate, using a simple example, the computation of MR curvature in an undirected graph. In this work, we have computed the average MR curvature of edges (MRE) in undirected financial networks by ignoring the edge weights and using equation (2.6).

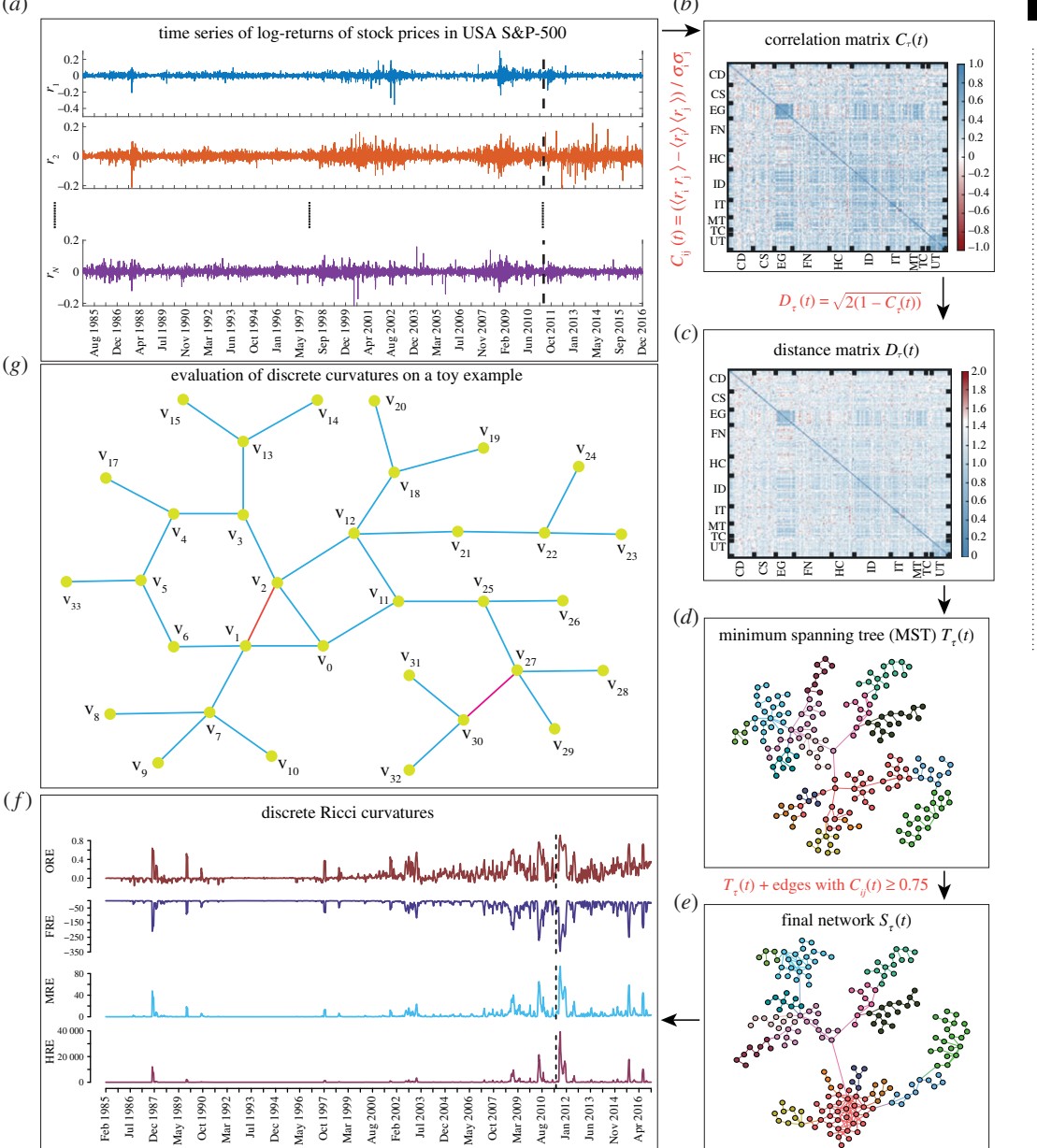

**Figure 1.** Schematic diagram describing the evaluation of discrete Ricci curvatures in correlation-based networks constructed from log-returns of USA S&P-500 market stocks. (*a*) Time series of log-returns over a 32-year period (1985–2016). (*b*) An arbitrarily chosen cross-correlation matrix $\boldsymbol{C}_\tau(t)$ for epoch ending on 4 May 2011. (*c*) Corresponding distance matrix $\boldsymbol{D}_\tau(t) = \sqrt{2(1 - \boldsymbol{C}_\tau(t))}$ used for the construction of the threshold network. (*d*) MST $T_\tau(t)$ constructed using the distance matrix $D_\tau(t)$. (*e*) Threshold network $S_\tau(t)$ constructed by adding edges with $C_{ij}(t) \geq 0.75$ to the MST $T_\tau(t)$. (*f*) Evolution of the average of four discrete Ricci curvatures for edges, namely, Ollivier–Ricci (ORE), Forman–Ricci (FRE), Menger–Ricci (MRE) and Haantjes–Ricci (HRE), computed using the threshold networks $S_\tau(t)$ constructed from correlation matrices over time epochs of $\tau = 22$ days and overlapping shift of $\Delta\tau = 5$ days. In this figure, $C_\tau(t)$, $D_\tau(t)$, $T_\tau(t)$ and $S_\tau(t)$ shown in (*b*)–(*e*) correspond to the correlation frame denoted by vertical dashed line in (*a*). (*g*) Evaluation of discrete Ricci curvatures on a toy example network which is undirected and unweighted. Here, the edge between $v_{27}$ and $v_{30}$ has a highly negative FR curvature as it depends on the degree of the two nodes or number of neighbouring edges. However, the edge between $v_{27}$ and $v_{30}$ has MR and HR curvature equal to zero as the edge under consideration is not part of any triangles or cycles, respectively. Moreover, the edge between $v_1$ and $v_2$ also has a highly negative FR curvature as the degree of both anchoring vertices is 4. By contrast, the edge between $v_1$ and $v_2$ has positive MR and HR curvature as the edge is part of a triangle which contributes to MR curvature and the edge is part of a triangle, a pentagon and a hexagon which contribute to HR curvature. For both the edges between $v_{27}$ and $v_{30}$ and between $v_1$ and $v_2$, one can compute OR curvature; however, only triangles, quadrangles and pentagons make positive contribution to the OR curvature in unweighted and undirected networks. Specifically, the edge between $v_1$ and $v_2$ is part of a triangle, a pentagon and a hexagon; however, only the triangle and pentagon make positive contribution to OR curvature.

## 2.4. Haantjes–Ricci curvature

We have also applied another notion of metric curvature to networks which is based on the suggestion of Finsler and was developed by his student Haantjes [69]. Haantjes defined the curvature of a metric curve as the ratio between the length of an arc of the curve and that of the chord it subtends. More precisely, given a curve $c$ in a metric space $(M, d)$, and given three points $p, q, r$ on $c$, $p$ between $q$ and $r$, the Haantjes curvature at the point $p$ is defined as

$$\kappa_H^2(p) = 24 \lim_{q,r \to p} \frac{l(\widehat{qr}) - d(q, r)}{(d(q, r))^3},$$ (2.7)

where $l(\widehat{qr})$ denotes the length, in the intrinsic metric induced by $d$, of the arc $\widehat{qr}$. In networks, $\widehat{qr}$ can be replaced by a path $\pi = v_0, v_1, \ldots, v_n$ between two nodes $v_0$ and $v_n$, and the subtending chord by the edge $e = (v_0, v_n)$ between the two nodes. Recently, some of us [55,68] have defined the Haantjes curvature of such a simple path $\pi$ as

$$\kappa_H^2(\pi) = \frac{l(\pi) - l(v_0, v_n)}{l(v_0, v_n)^3},$$ (2.8)

where, if the graph is a metric graph, $l(v_0, v_n) = d(v_0, v_n)$, that is the shortest path distance between nodes $v_0$ and $v_n$. In particular, for the combinatorial metric (or unweighted graphs), we obtain that $\kappa_H(\pi) = \sqrt{n-1}$, where $\pi = v_0, v_1, \ldots, v_n$ is as above. Note that considering simple paths in graphs concords with the classical definition of Haantjes curvature, since a metric arc is, by its very definition, a simple curve. Thereafter, the Haantjes–Ricci (HR) curvature of an edge $e$ [55,68] can be defined as

$$\kappa_H(e) = \sum_{\pi \sim e} \kappa_H(\pi),$$ (2.9)

where $\pi \sim e$ denote the paths that connect the nodes anchoring the edge $e$. Note that while MR curvature considers only triangles or simple paths of length 2 between two nodes anchoring an edge in unweighted graphs, the HR curvature considers even longer paths between the same two nodes anchoring an edge (figure 1). Moreover, for triangles endowed with the combinatorial metric, the two notions by Menger and Haantjes coincide, up to a universal constant. In electronic supplementary material, figure S4, we illustrate, using a simple example, the computation of HR curvature in an undirected graph. In this work, we have computed the average HR curvature of edges (HRE) in undirected financial networks by ignoring the edge weights and using equation (2.9). Moreover, due to computational constraints, we only consider simple paths $\pi$ of length less than or equal to 4 between the two vertices at the ends of any edge while computing its HR curvature using equation (2.9) in analysed networks. Note that both Menger and Haantjes curvature are positive in undirected networks, and they capture the (absolute value of) geodesics dispersal rate of the classical Ricci curvature.

# 3. Data and methods

## 3.1. Data description

The data were collected from the public domain of Yahoo finance database [70] for two markets: USA S&P-500 index and Japanese Nikkei-225 index. The investigation in this work spans a 32-year period from 2 January 1985 to 30 December 2016. We analysed the daily closure price data of $N = 194$ stocks for $T = 8068$ days for USA S&P-500 and $N = 165$ stocks for $T = 7998$ days for Japanese Nikkei-225 markets. Electronic supplementary material, tables S1 and S2 give the lists of 194 and 165 stocks (along with their sectors) for the USA S&P-500 and Japanese Nikkei-225 markets, respectively, and these stocks are present in the two markets for the entire 32-year period considered here.

## 3.2. Cross-correlation and distance matrices

We present a study of time evolution of the cross-correlation structures of return time series for $N$ stocks (figure 1). The daily return time series is constructed as $r_k(t) = \ln P_k(t) - \ln P_k(t-1)$, where $P_k(t)$ is the adjusted closing price of the $k$th stock at time $t$ (trading day). Then, the cross-correlation matrix is

constructed using equal-time Pearson cross-correlation coefficients,

$$C_{ij}(t) = \frac{(\langle r_i r_j \rangle - \langle r_i \rangle \langle r_j \rangle)}{\sigma_i \sigma_j},$$

where $i, j = 1, \ldots, N$, $t$ indicates the end date of the epoch of size $\tau$ days, and the means $\langle \cdots \rangle$ as well as the standard deviations $\sigma_k$ are computed over that epoch.

Instead of working with the correlation coefficient $C_{ij}(t)$, we use the 'ultrametric' distance measure:

$$D_{ij}(t) = \sqrt{2(1 - C_{ij}(t))},$$

such that $0 \leq D_{ij}(t) \leq 2$, which can be used for the construction of networks [11,12,18,27].

Here, we computed daily return cross-correlation matrix $\mathbf{C}_\tau(t)$ over the short epoch of $\tau = 22$ days and shift of the rolling epoch by $\Delta\tau = 5$ days, for (i) $N = 194$ stocks of USA S&P-500 for a return series of $T = 8068$ days, and (ii) $N = 165$ stocks of Japan Nikkei-225 for $T = 7998$ days, during the 32-year period from 1985 to 2016. We use epochs of $\tau = 22$ days (one trading month) to obtain a balance between choosing short epochs for detecting changes and long ones for reducing fluctuations. In the main text, we show results for networks constructed from correlation matrices with overlapping epochs of $\Delta\tau = 5$ days, while in the electronic supplementary material, we show results for networks constructed from correlation matrices with non-overlapping epochs of $\Delta\tau = 22$ days.

## 3.3. Network construction

For a given time epoch of $\tau$ days ending on trading day $t$, the distance matrix $\mathbf{D}_\tau(t)$ constructed from the correlation matrix between the 194 stocks in USA S&P-500 index or the 165 stocks in Japan Nikkei-225 index can be viewed as an undirected complete graph $G_\tau(t)$, where the weight of an edge between stocks $i$ and $j$ is given by the distance $D_{ij}(t)$. For the time epoch of $\tau$ days ending on trading day $t$, we start with this edge-weighted complete graph $G_\tau(t)$ and create the MST $T_\tau(t)$ using Prim's algorithm [71]. Thereafter, we add edges in $G_\tau(t)$ with $C_{ij}(t) \geq 0.75$ to $T_\tau(t)$ to obtain the graph $S_\tau(t)$ (figure 1). We will use the graph $S_\tau(t)$ to compute different discrete Ricci curvatures and other network measures. We remark that the procedure used here to construct the graph $S_\tau(t)$ follows previous works [18,50] on analysis of correlation-based networks of stock markets.

Intuitively, the motivation behind the above method of graph construction can be understood as follows. Firstly, the MST method gives a connected (spanning) graph between all nodes (stocks) in the specific market. Secondly, the addition of edges between nodes (stocks) with correlation $C_{ij}(t) \geq 0.75$ ensures that the important edges are also captured in the graph $S_\tau(t)$.

## 3.4. Common network measures

Given an undirected graph $G(V, E)$ with the sets of vertices or nodes $V$ and edges $E$, the number of edges is given by the cardinality of set $E$, that is $m = |E|$, and the number of nodes is given by the cardinality of set $V$, that is $n = |V|$. The edge density of such a graph is given by the ratio of the number of edges $m$ divided by the number of possible edges, that is, $2m/n(n-1)$. The average degree $\langle k \rangle$ of the graph gives the average number of edges per node, that is, $\langle k \rangle = m/n$. In the case of an edge-weighted graph where $a_{ij}$ denotes the weight of the edge between nodes $i$ and $j$, one can also compute its average weighted degree $\langle k_w \rangle$ which gives the average of the sum of the weights of the edges connected to nodes, that is, $\langle k_w \rangle = \frac{m_w}{n}$ where $m_w = \sum_{i,j \in V} a_{ij}$. For any pair of nodes $i$ and $j$ in the graph, one can compute the shortest path length $d_{ij}$ between them. Thereafter, the average shortest path length $\langle L \rangle$ is given by the average of the shortest path lengths between all pairs of nodes in the graph, that is,

$$\langle L \rangle = \frac{1}{n(n-1)} \sum_{i \neq j \in V} d_{ij}.$$

The diameter is given by the maximum of the shortest paths between all pairs of nodes in the graph, i.e. $\max\{d_{ij} \ \forall i, j \in V\}$. The communication efficiency [72] of a graph is an indicator of its global ability to exchange information across the network. The communication efficiency $CE$ of a graph is given by

$$CE = \frac{1}{n(n-1)} \sum_{i \neq j \in V} \frac{1}{d_{ij}}.$$

Modularity measures the extent of community structure in the network and community detection algorithms aim to partition the graph into communities such that the modularity $Q$ attains the maximum value [73]. The modularity $Q$ is given by the equation [73,74]

$$Q = \frac{1}{2m_w} \sum_{i \neq j \in V} \left[ a_{ij} - \frac{k_i k_j}{2m_w} \right] \delta(c_i, c_j),$$

where $k_i$ and $k_j$ give the sum of weights of edges attached to nodes $i$ and $j$, respectively, $c_i$ and $c_j$ give the communities of $i$ and $j$, respectively, and $\delta(c_i, c_j)$ is equal to 1 if $c_i = c_j$ else 0. Here, we use Louvain method [74] to compute the modularity of the edge-weighted networks. Network entropy is an average measure of graph heterogeneity as it quantifies the diversity of edge distribution using the remaining degree distribution $q_k$ [75]. $q_k$ denotes the probability of a node to have remaining (excess) degree $k$ and is given by $q_k = (k+1)p_{k+1}/\langle k \rangle$ where $p_{k+1}$ denotes the probability of a node to have degree $k+1$. The network entropy $H(q)$ of a graph is then given by

$$H(q) = - \sum_k q_k log(q_k).$$

The above-mentioned network measures were computed in stock market networks using the Python package `NetworkX` [76].

## 3.5. GARCH($p, q$) process

The generalized ARCH process GARCH($p, q$) was introduced by Bollerslev [77]. The variable $x_t$, a strong white noise process, can be written in terms of a time-dependent standard deviation $\sigma_t$, such that $x_t \equiv \eta_t \sigma_t$, where $\eta_t$ is a random Gaussian process with zero mean and unit variance.

The simplest GARCH process is the GARCH(1,1) process, with Gaussian conditional probability distribution function

$$\sigma_t^2 = \alpha_0 + \alpha_1 x_{t-1}^2 + \beta_1 \sigma_{t-1}^2, \tag{3.1}$$

where $\alpha_0 > 0$ and $\alpha_1 \geq 0$; $\beta_1$ is an additional control parameter. One can rewrite equation (3.1) as a random multiplicative process

$$\sigma_t^2 = \alpha_0 + (\alpha_1 \eta_{t-1}^2 + \beta_1)\sigma_{t-1}^2. \tag{3.2}$$

For calculating this we have used an in-built function from Matlab garch (https://in.mathworks.com/help/econ/garch.html).

## 3.6. Minimum risk portfolio

We calculated the minimum risk portfolio in the Markowitz framework, as a measure of risk-aversion of each investor with maximized expected returns and minimized variance. In this model, the variance of a portfolio shows the importance of effective diversification of investments to minimize the total risk of a portfolio. The Markowitz model minimizes $w'\Omega w - \phi R'w$ with respect to the normalized weight vector $w$, where $\Omega$ is the covariance matrix calculated from the stock log-returns, $\phi$ is the measure of risk appetite of investor and $R'$ is the expected return of the assets. We set short-selling constraint, $\phi = 0$ and $w_i \geq 0$ which entails a convex combination of stock return for finding the minimum risk portfolio. For calculating this we have used an in-built function from Matlab (https://in.mathworks.com/help/finance/portfolio.html).

# 4. Results and discussion

We analyse here the time series of the logarithmic returns of the stocks in the USA S&P-500 and Japanese Nikkei-225 markets over a period of 32 years (1985–2016) by constructing the corresponding Pearson cross-correlation matrices $C_\tau(t)$. We then use cross-correlation matrices $C_\tau(t)$ computed over time epochs of size $\tau = 22$ days with either overlapping or non-overlapping epochs (i.e. shifts of $\Delta\tau = 5$ or 22 days, respectively, and ending on trading days $t$) to study the evolution of the correlation-based networks $S_\tau(t)$ and corresponding network properties, especially edge-centric geometric measures. Figure 1 gives an overview of our evaluation of discrete Ricci curvatures in correlation-based

threshold networks constructed from log-returns of market stocks. Figure 1a shows the daily log-returns over the 32-year period (1985–2016). An arbitrarily chosen cross-correlation matrix $C_\tau(t)$ over time epoch of $\tau = 22$ days and $\Delta\tau = 5$ days ending on 4 May 2011 and corresponding distance matrix $D_\tau(t) = \sqrt{2(1 - C_\tau(t))}$ are shown in figure 1b,c, respectively. The MST $T_\tau(t)$ constructed from the distance matrix $D_\tau(t)$ is shown in figure 1d. Thereafter, a threshold network $S_\tau(t)$ is constructed using MST $T_\tau(t)$ and edges with $C_{ij}(t) \geq 0.75$, as shown in figure 1e. The discrete Ricci curvatures are computed from the threshold networks. In figure 1f, we show the evolution of the discrete curvatures in threshold networks over the 32-year period. In figure 1g, we motivate the four discrete Ricci curvatures considered here using a simple example network.

A major goal of this research is to evaluate different notions of discrete Ricci curvature for their ability to unravel the structure of complex financial networks and serve as indicators of market instabilities. Previously, Sandhu et al. [50] have analysed the USA S&P-500 market over a period of 15 years (1998–2013) to show that the average Ollivier–Ricci (OR) curvature of edges (ORE) in threshold networks increases during periods of financial crisis. Here, we extend the analysis by Sandhu et al. [50] to (i) two different stock markets, namely, USA S&P-500 and Japanese Nikkei-225, (ii) a span of 32 years (1985–2016), (iii) four traditional market indicators (namely, index log-returns $r$, mean market correlation $\mu$, volatility of the market index $r$ estimated using GARCH(1,1) process, and risk $\sigma_P$ corresponding to the minimum risk Markowitz portfolio of all the stocks in the market), and (iv) four notions of discrete Ricci curvature for networks. Since discretizations of Ricci curvature are unable to capture the entire properties of the classical Ricci curvature defined on continuous spaces, the four discrete Ricci curvatures evaluated here can shed light on different properties of analysed networks [48]. In particular, some of us have introduced another discretization, Forman–Ricci (FR) curvature, to the domain of networks [43]. Note that OR curvature captures the volume growth property of classical Ricci curvature while FR curvature captures the geodesic dispersal property [48]. Nevertheless, our empirical analysis has shown that the two discrete notions, OR and FR curvature, are highly correlated in model and real-world networks [48]. Importantly, in large networks, computation of the OR curvature is intensive while that of the FR curvature is simple as the latter depends only on immediate neighbours of an edge [48]. Therefore, we started by investigating the ability of FR curvature to capture the structure of complex financial networks.

Figure 2 shows the comparisons of threshold networks, as well as the behaviour of index log-returns $r$ and average FR curvature of edges (FRE), for (a) bubble and (b) crash periods, of the USA S&P-500 market. The upper panel of figure 2a shows the threshold networks near the US Housing bubble period (2006–2007) at four distinct epochs of $\tau = 22$ days ending on trading days $t$ equal to 23 January 2006, 10 May 2006, 29 June 2006 and 6 November 2006, with threshold $C_{ij}(t) \geq 0.75$. Number of edges and communities in these four threshold networks are 251, 220, 996, 220 and 13, 16, 11, 14, respectively. The colour of the nodes correspond to the different communities determined by Louvain method [74] for community detection. The plots of log-returns of S&P-500 index $r$ (blue colour line) and FRE (sienna colour line) around the US Housing bubble period are shown in the lower panel of figure 2a. Threshold networks show higher number (996) of edges and lower number (11) of communities for high (negative) values of FRE, but there is not much variation of $r$. In electronic supplementary material, figure S5, we show that the FRE captures the same features for three other thresholds $C_{ij}(t) \geq 0.55$, $C_{ij}(t) \geq 0.65$ and $C_{ij}(t) \geq 0.85$, and the numbers of edges and communities for each threshold are listed in electronic supplementary material, table S3. The measure FRE is sensitive to both local (sectoral) and global (market) fluctuations, and shows a local minimum during bubble. Note that during a bubble, only a few sectors of the market perform well compared to the others (the stocks within the well-performing sectors are highly correlated, but the inter-sectoral correlations are low). It is hard to identify bubble by only monitoring the market index as the returns do not show much volatility. Figure 2b shows the same for the period around the August 2011 stock markets fall at four distinct epochs of $\tau = 22$ days ending on trading days $t$ equal to 7 January 2011, 4 May 2011, 2 September 2011 and 3 February 2012, with threshold $C_{ij}(t) \geq 0.75$. Number of edges and communities in these four threshold networks are 197, 245, 16 004, 198 and 14, 16, 4, 15, respectively. During the crash, the threshold network shows sufficiently higher number of edges and extremely low number of communities. In electronic supplementary material, figure S6, we show that the FRE captures the same features for three other thresholds $C_{ij}(t) \geq 0.55$, $C_{ij}(t) \geq 0.65$ and $C_{ij}(t) \geq 0.85$, and the numbers of edges and communities for each threshold is listed in electronic supplementary material, table S3. The plots of log-returns $r$ of S&P-500 index (blue colour line) and FRE (sienna colour line) are shown around the August 2011 stock markets fall period in the lower panel of figure 2b. Note that during a market crash $r$ displays high volatility and FRE shows a significant decrease (local

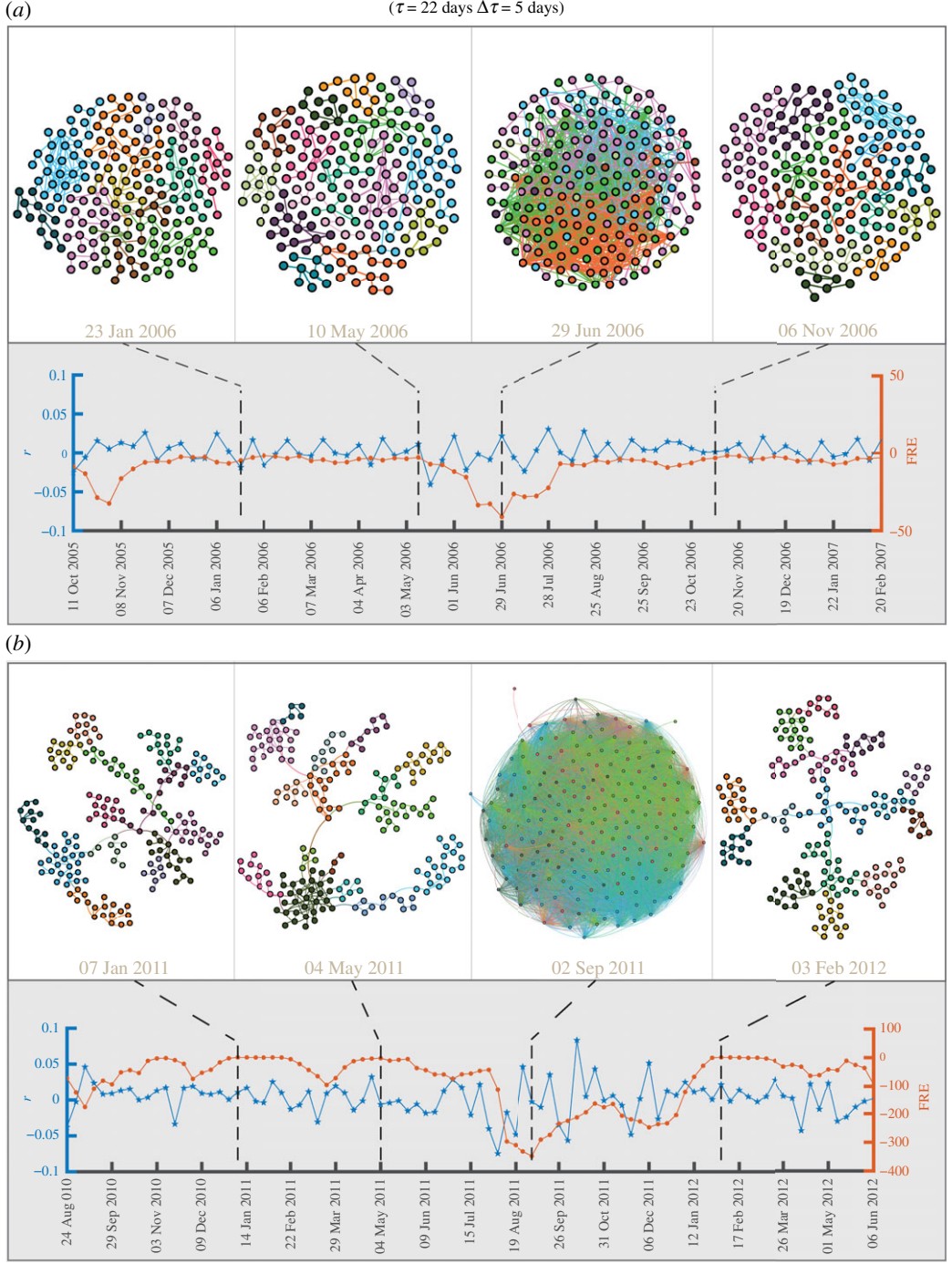

**Figure 2.** (a) (Upper panel) Visualization of threshold networks for USA S&P-500 market around the US Housing bubble period (2006–2007) at four distinct epochs of $\tau = 22$ days ending on trading days 23 January 2006, 10 May 2006, 29 June 2006 and 6 November 2006, with threshold $C_{ij}(t) \geq 0.75$. Here, the colour of the nodes corresponds to the different communities determined by Louvain method for community detection. Threshold networks show higher number of edges and lower number of communities during a bubble. (Lower panel) Plot shows the evolution of log-returns $r$ of S&P-500 index (blue colour line) and average Forman–Ricci curvature of edges (FRE) (sienna colour line) for the period around the US Housing bubble. The FRE measure, constructed from threshold networks, is sensitive to both local (sectoral) and global fluctuations of the market, and shows a local minimum (more negative) during the bubble, whereas not much variation is seen in $r$ (low volatility). (b) (Upper panel) Visualization of threshold networks for USA S&P-500 market around the August 2011 stock markets fall at four distinct epochs of $\tau = 22$ days ending on 7 January 2011, 4 May 2011, 2 September 2011 and 3 February 2012 with threshold $C_{ij}(t) \geq 0.75$. Here, the threshold network shows significantly higher number of edges and lower number of communities during the crash. (Lower panel) Plot shows the evolution of log-returns $r$ of S&P 500 index (blue colour line) and FRE (sienna colour line) for the period around the August 2011 stock markets fall. During the crash $r$ has high fluctuations (high volatility) and FRE decreases significantly (local minima).

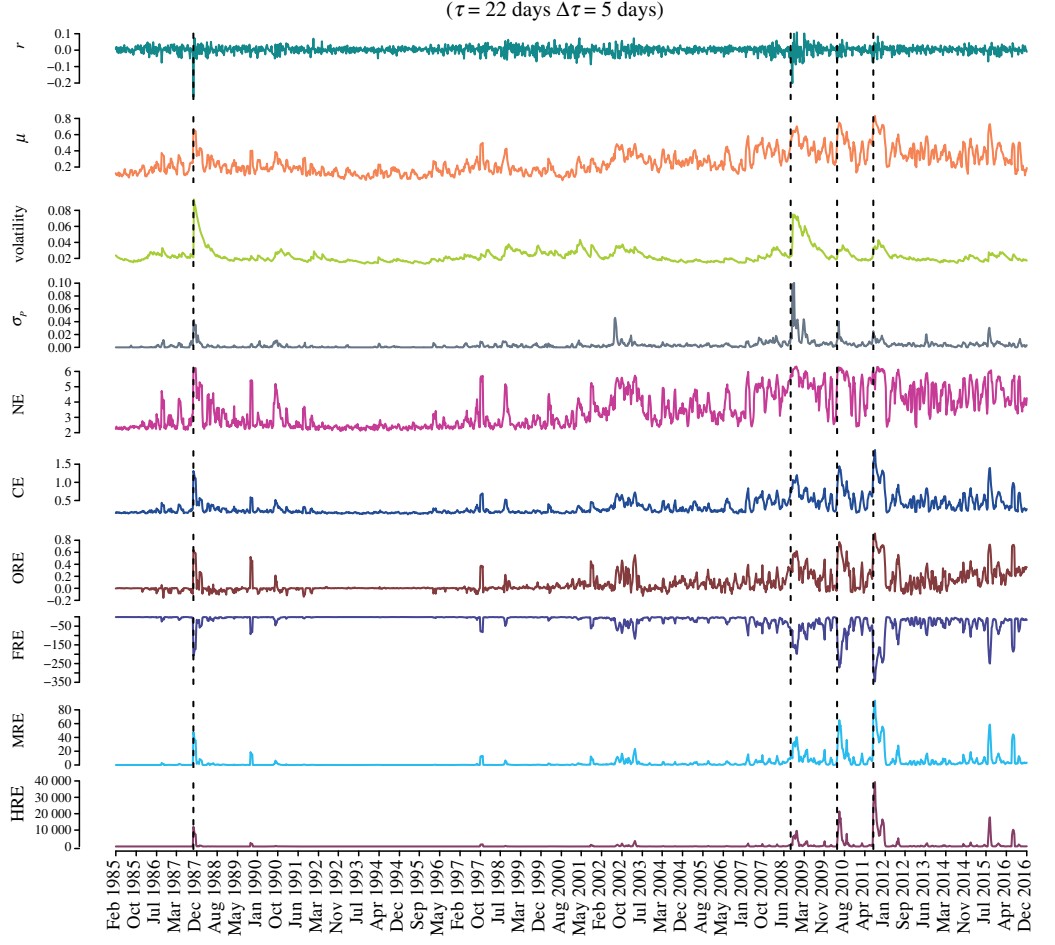

**Figure 3.** Evolution of the market indicators and edge-centric geometric curvatures for the USA S&P-500 market. From top to bottom, we plot the index log-returns $r$, mean market correlation $\mu$, volatility of the market index $r$ estimated using GARCH(1,1) process, risk $\sigma_P$ corresponding to the minimum risk Markowitz portfolio of all the stocks in the market, network entropy (NE), communication efficiency (CE), average of Ollivier–Ricci (ORE), Forman–Ricci (FRE), Menger–Ricci (MRE) and Haantjes–Ricci (HRE) curvature of edges evaluated from the correlation matrices $\boldsymbol{C}_\tau(t)$ of epoch size $\tau = 22$ days and an overlapping shift of $\Delta\tau = 5$ days. Four vertical dashed lines indicate the epochs of four important crashes: Black Monday 1987, Lehman Brothers crash 2008, DJ Flash crash 2010 and August 2011 stock markets fall (table 1).

minimum). Earlier Sandhu *et al.* [50] had focused on OR curvature as an indicator of crashes. Here, we additionally show that discrete Ricci curvatures, especially FR curvature, are sensitive and can detect both crash (market volatility high) and bubble (market volatility low).

It is often difficult to gauge the state of the market by simply monitoring the market index or its log-returns. There exist no simple definitions of a market crash or a market bubble. The market becomes extremely correlated and volatile during a crash, but a bubble is even harder to detect as the volatility is relatively low and only certain sectors perform very well (stocks show high correlation) but the rest of the market behaves like normal or 'business-as-usual'. Traditionally, the volatility of the market captures the 'fear' and the evaluated risk captures the 'fragility' of the market. Some of us showed in our earlier papers that the mean market correlation and the spectral properties of the cross-correlation matrices can be used to study the market states [20] and identify the precursors of market instabilities [22]. A goal of this study is to show that the state of the market can be continuously monitored with certain network-based measures. Thus, we next performed a comparative investigation of several network measures, especially, the four discrete notions of Ricci curvature.

Figures 3 and 4 show for USA S&P-500 market and Japanese Nikkei-225 market, respectively, the temporal evolution of the market indicators and network measures, mainly edge-centric Ricci curvatures computed from the correlation matrices $\boldsymbol{C}_\tau(t)$ of epoch size $\tau = 22$ days and overlapping shift of $\Delta\tau = 5$ days, over a 32-year period (1985–2016). From top to bottom, the plots represent index

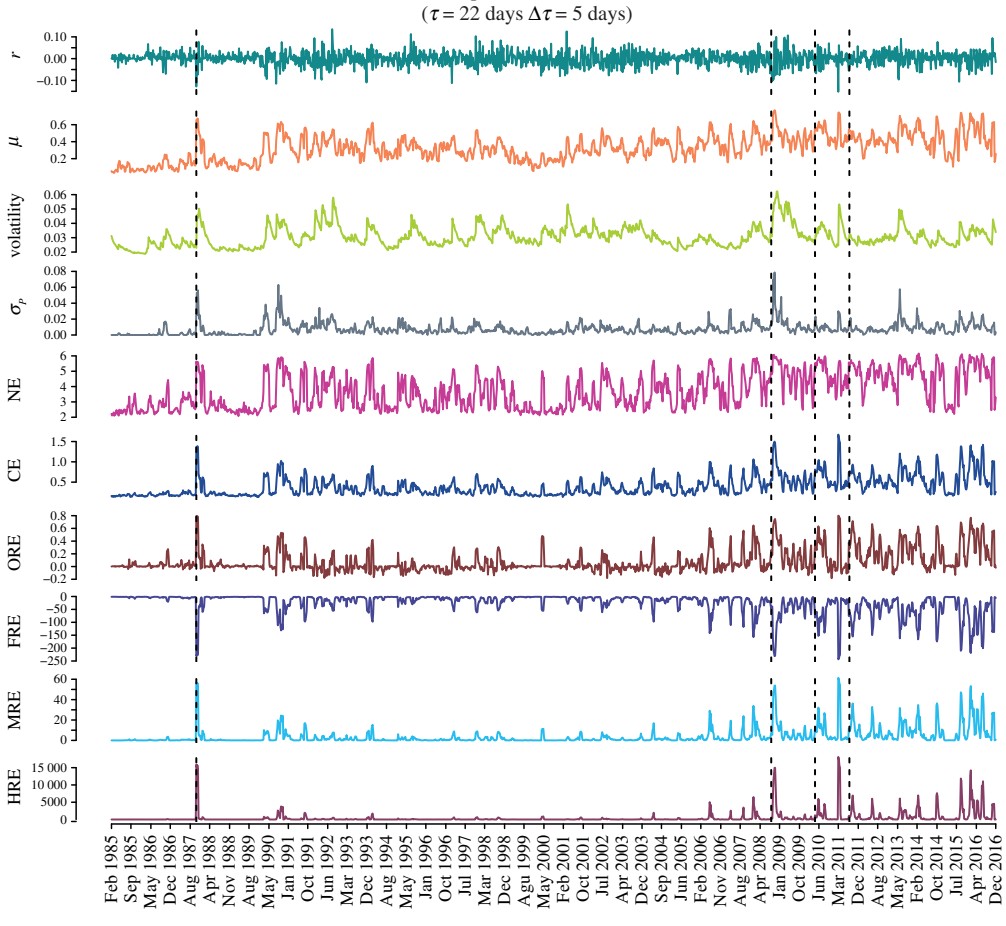

**Figure 4.** Evolution of the market indicators and edge-centric geometric curvatures for the Japanese Nikkei-225 market. From top to bottom, we plot the index log-returns $r$, mean market correlation $\mu$, volatility of the market index $r$ estimated using GARCH(1,1) process, risk $\sigma_P$ corresponding to the minimum risk Markowitz portfolio of all the stocks in the market, network entropy (NE), communication efficiency (CE), average of Ollivier–Ricci (ORE), Forman–Ricci (FRE), Menger–Ricci (MRE) and Haantjes–Ricci (HRE) curvature of edges evaluated from the correlation matrices $C_\tau(t)$ of epoch size $\tau = 22$ days and an overlapping shift of $\Delta\tau = 5$ days. Four vertical dashed lines indicate the epochs of four important crashes: Black Monday 1987, Lehman Brothers crash 2008, DJ Flash crash 2010, and August 2011 stock markets fall (table 1).

**Table 1.** List of major crashes and bubbles in stock markets of USA and Japan between 1985 and 2016 [78–83].

| serial number | major crashes and bubbles | period | affected region |
|---|---|---|---|
| 1 | Black Monday | 19 Oct 1987 | USA, Japan |
| 2 | Friday the 13th mini crash | 13 Oct 1989 | USA |
| 3 | Early 90s recession | 1990 | USA |
| 4 | Mini crash due to Asian financial crisis | 27 Oct 1997 | USA |
| 5 | Lost decade | 2001–2010 | Japan |
| 6 | 9/11 financial crisis | 11 Sep 2001 | USA, Japan |
| 7 | Stock market downturn of 2002 | 9 Oct 2002 | USA, Japan |
| 8 | US Housing bubble | 2005–2007 | USA |
| 9 | Lehman Brothers crash | 16 Sep 2008 | USA, Japan |
| 10 | Dow Jones (DJ) Flash crash | 6 May 2010 | USA, Japan |
| 11 | Tsunami and Fukushima disaster | 11 Mar 2011 | Japan |
| 12 | August 2011 stock markets fall | 8 Aug 2011 | USA, Japan |
| 13 | Chinese Black Monday and 2015–2016 sell off | 24 Aug 2015 | USA |

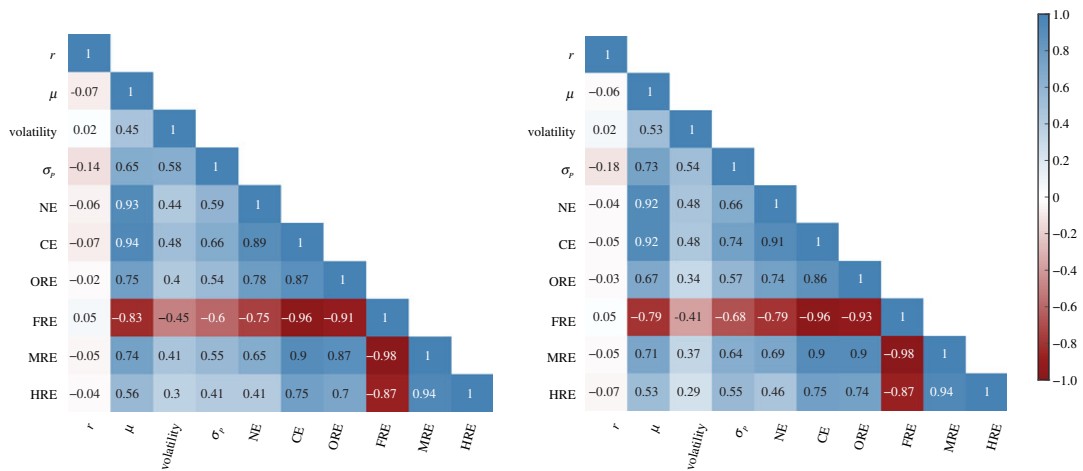

**Figure 5.** Correlogram plots of (*a*) USA S&P-500 and (*b*) Japan Nikkei-225 markets, for the traditional market indicators (index returns *r*, mean market correlation $\mu$, volatility, and minimum risk portfolio $\sigma_P$), global network properties (network entropy NE and communication efficiency CE) and discrete Ricci curvatures for edges (ORE, FRE, MRE and HRE), computed for epochs of size $\tau = 22$ days and overlapping shift $\Delta\tau = 5$ days.

log-returns *r*, mean market correlation $\mu$, volatility of the market index *r* estimated using GARCH(1,1) process, risk $\sigma_P$ corresponding to the minimum risk Markowitz portfolio of all the stocks in the market, network entropy (NE), communication efficiency (CE), average of OR, FR, MR and HR curvature of edges. We find that the four Ricci-type curvatures, namely, ORE, FRE, MRE and HRE, along with the other important indicators of the markets, viz., the log-returns *r*, volatility, minimum risk $\sigma_P$ and mean market correlation $\mu$, are excellent indicators of market instabilities (bubbles and crashes). We highlight that the four discrete Ricci curvatures can capture important crashes and bubbles listed in table 1 in the two markets during the 32-year period studied here.

In electronic supplementary material, figure S7, we show the temporal evolution of the four discrete Ricci curvatures computed in threshold networks $S_\tau(t)$ obtained using three different thresholds, $C_{ij}(t) \geq 0.65$ (cyan colour), $C_{ij}(t) \geq 0.75$ (dark blue colour) and $C_{ij}(t) \geq 0.85$ (sienna colour), for the two markets. It is seen that the absolute value of ORE, FRE, MRE and HRE decreases with the increase in the threshold $C_{ij}(t)$ used to construct $S_\tau(t)$. Regardless of the three thresholds used to construct the threshold networks $S_\tau(t)$, we show that the four discrete Ricci curvatures are fine indicators of market instabilities.

In previous work, Sandhu *et al.* [50] had contrasted the temporal evolution of ORE in threshold networks for USA S&P-500 market with NE, graph diameter and average shortest path length. Here, we have studied the temporal evolution of a larger set of network measures in threshold networks for USA S&P-500 and Japanese Nikkei-225 markets computed from the correlation matrices $C_\tau(t)$ of epoch size $\tau = 22$ days and overlapping shift of $\Delta\tau = 5$ days, over a 32-year period (1985–2016). From figures 3 and 4, it is seen that NE and CE are also excellent indicators of market instabilities. In fact, we find that common network measures such as number of edges, edge density, average degree, average shortest path length, graph diameter, average clustering coefficient and modularity are also good indicators of market instabilities (electronic supplementary material, figure S8).

In electronic supplementary material, figures S9 and S10, we show the temporal evolution of the market indicators and several network measures (including edge-centric Ricci curvatures) computed from the correlation matrices $C_\tau(t)$ of epoch size $\tau = 22$ days and non-overlapping shift of $\Delta\tau = 22$ days, over a 32-year period (1985–2016) in the two markets. It can be seen that our results are also not dependent on the choice of overlapping or non-overlapping shift used to construct the cross-correlation matrices and threshold networks.

Figure 5 shows the correlogram plots of (*a*) USA S&P-500 and (*b*) Japanese Nikkei-225 markets, for the traditional market indicators (index returns *r*, mean market correlation $\mu$, volatility, and minimum portfolio risk $\sigma_P$), network properties (NE and CE) and discrete Ricci curvatures (ORE, FRE, MRE and HRE), computed for epoch size $\tau = 22$ days and overlapping shift of $\Delta\tau = 5$ days. In electronic supplementary material, figure S11, we show the correlogram plots for the traditional market indicators and network properties including discrete Ricci curvatures computed for epoch size $\tau = 22$ days and non-overlapping shift of $\Delta\tau = 22$ days in the two markets. Notably, FRE shows the highest

correlation among the four discrete Ricci curvatures with the four traditional market indicators in the two markets, and thus, FRE is an excellent indicator for market risk that captures local to global system-level fragility of the markets. Furthermore, both NE and CE also have high correlation with the four traditional market indicators. Therefore, these measures can be used to monitor the health of the financial system and forecast market crashes or downturns. Overall, we show that FRE is a simple yet powerful tool for capturing the correlation structure of a dynamically changing network.

## 5. Conclusion

In this paper, we have employed geometry-inspired network curvature measures to characterize the correlation structures of the financial systems and used them as generic indicators for detecting market instabilities (bubbles and crashes). We reiterate here that it is often difficult to gauge the state of the market by simply monitoring the market index or its log-returns. There exist no simple definitions of a market crash or a market bubble. The market becomes extremely correlated and volatile during a crash, but a bubble is even harder to detect as the volatility is relatively low and only certain sectors perform very well (stocks show high correlation) but the rest of the market behaves like normal or 'business-as-usual'. We have examined the daily returns from a set of stocks comprising the USA S&P-500 and the Japanese Nikkei-225 over a 32-year period, and monitored the changes in the edge-centric geometric curvatures. Our results are very robust as we have studied two very different markets, and for a very long period of 32 years with several interesting market events (bubbles and crashes; table 1). We showed that the results are not very sensitive to the choice of overlapping or non-overlapping epochs used to construct the cross-correlation matrices and threshold networks (figures 3–4; electronic supplementary material, figures S8–S10). Further, the choice of the thresholds for constructing networks also has little influence on their behaviour as indicators (electronic supplementary material, figures S5–S7). In addition, to test the robustness of our methodology in the current paper, we have added small amounts of Gaussian noise to the empirical correlation matrices for the USA S&P-500 market, and reproduced the evolution of the topological properties as well as the geometric curvature measures over the 32-year period. Specifically, we have found that the results are not sensitive to small amounts of noise or random reshuffling of data (electronic supplementary material, figure S12). We found that the four different notions of discrete Ricci curvature captured well the system-level features of the market and hence we were able to distinguish between the normal or 'business-as-usual' periods and all the major market crises (bubbles and crashes) using the network-centric indicators. Our studies confirmed that during a normal period the market is very modular and heterogeneous, whereas during an instability (crisis) the market is more homogeneous, highly connected and less modular.

Interestingly, our methodology picks up many peaks other than the major crashes and bubbles; these are neither spurious nor false positives. Unlike the major crashes and bubbles which are well-documented in the financial literature (or listed in Internet sources, see table 1), many of these peaks correspond to interesting events that are not well understood or recorded in the literature. In fact the financial markets are often driven by endogenous and exogenous factors. Moreover, there are often multiple reasons leading to a market crash or a bubble burst. The study and characterization of such market events, including exogenous shocks, bubble bursts and anomalies, corresponding to such peaks has already been done in our earlier papers [20–22,30]. The findings of the present paper are in concordance with the earlier ones.

It is important to note that partial correlations can detect direct as opposed to plausibly indirect connections among components of the stock market. In the Econophysics literature (e.g. [30,84–87]), researchers have used partial correlations for analysing the dynamics and constructing networks of stock markets. Partial correlations are particularly relevant when people study eigenvalue spectra (market, group and random modes), or network centrality measures, by first filtering out the spurious correlations. However, it has been observed [30,86] that partial correlations are less successful in picking the cluster or group dynamics, and the networks arising from partial correlations are also less stable. In this contribution, we are more interested in the market indicators and the use of discrete Ricci curvatures as generic indicators, for which we prefer to work with the more stable correlation matrices.

Also, we find from these geometric measures that there are succinct and inherent differences in the two markets, USA S&P-500 and Japan Nikkei-225. Importantly, among four Ricci-type curvature measures, the Forman–Ricci curvature of edges (FRE) correlates highest with the traditional market

indicators and acts as an excellent indicator for the system-level fear (volatility) and fragility (risk) for both the markets. These new insights may help us in future to better understand tipping points, systemic risk and resilience in financial networks, and enable us to develop monitoring tools required for the highly interconnected financial systems and perhaps forecast future financial crises and market slowdowns. These can be further generalized to study other economic systems, and may thus enable us to understand the highly complex and interconnected economic-financial systems.

Ethics. In this manuscript, there are no data or topics that need ethical approval.

Data accessibility. All data used are openly available for download on the websites of the relevant sources mentioned in the text and stated in the references section. All relevant data and codes for this study have been uploaded and made publicly available via the GitHub repository: https://github.com/asamallab/StockMarkNetIndicator.

Authors' contributions. A.S. and A.C. designed research; A.S., H.K.P., S.J.R., H.K., E.S., J.J. and A.C. performed research and analysed data; A.S., H.K.P. and S.J.R. prepared the figures; A.S. and A.C. supervised the research; A.S., E.S., J.J. and A.C. wrote the manuscript with input from the other authors. All authors have read and approved the manuscript.

Competing interests. We declare we have no competing interests.

Funding. A.S. acknowledges financial support from Max Planck Society Germany through the award of a Max Planck Partner Group in Mathematical Biology. E.S. and J.J. acknowledge support from the German-Israeli Foundation (GIF) grant no. I-1514-304.6/2019.

Acknowledgements. H.K.P. is grateful for financial support provided by UNAM-DGAPA and CONACYT Proyecto Fronteras 952. A.C. and H.K.P. acknowledge support from the projects UNAM-DGAPA-PAPIIT AG100819 and IN113620, and CONACyT Project Fronteras 201.

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
