## [Peer Review File · Royal Society Open Science]

Review History

RSOS-201734.R0 (Original submission)

Review form: Reviewer 1

Is the manuscript scientifically sound in its present form?

Yes

Are the interpretations and conclusions justified by the results?

Yes

Is the language acceptable?

Yes

Do you have any ethical concerns with this paper?

No

Have you any concerns about statistical analyses in this paper?

No

Recommendation?

Reject

Comments to the Author(s)

It is known that geometrical measures and space-curvature are associated with network properties and the relation between correlation networks MSTs and other information filtering networks and the space curvature has been investigated previously [see Complex networks on hyperbolic surfaces Physica A 2005 Volume 346 20-26].

The fact that Ricci curvature is related with other network measures is not a surprise neither a novelty.

The question the authors should have addressed is whether it is useful to map simpler network topological properties into more complicated geometrical properties.

This paper fails to prove that there are insights in the Ricci curvature properties that cannot be uncovered with traditional network topological properties. Furthermore, the interpretation in terms of space-curvature does not provide any simpler or more intuitive perspective and therefore it results in a useless exercise.

Review form: Reviewer 2**Is the manuscript scientifically sound in its present form?**

Yes

Are the interpretations and conclusions justified by the results?

Yes

Is the language acceptable?

Yes

Do you have any ethical concerns with this paper?

No

Have you any concerns about statistical analyses in this paper?

Yes

Recommendation?

Accept with minor revision (please list in comments)

Comments to the Author(s)

This is a very interesting study. The authors put forward and apply advanced geometrical methods involving different curvatures to characterize financial markets. As these are nonstationary with an often abrupt dynamics, development of new methods can be a rewarding enterprise. In the present case, I am sure that the guiding ideas are an important novel input facilitating the analysis of data from various complex systems, including financial markets.

The manuscript certainly contains enough results to warrant publication in Royal Society Open Science.

There are a few points which the authors should address before publication:

research:

+ Minimum spanning trees are notoriously known for lack of robustness. Put differently: if I change the data a little bit (playing here, shaking there) I often get very different MST. Has this been taken into account? What is the impact on the results presented? Are they sensitive to the particular details of the MST? I do not find the present discussion sufficient. The authors showed that the "results are not very sensitive to the choice of overlapping or non-overlapping windows". Good! But this is not the same point.

+ In figs 3 and 4 the different measures show several peaks. In some cases they are related to drastic events in the markets, as indicated by dashed lines. Much more should be said on the other peaks. Are they spurious? If so, why? To which extent does the presence of these peaks with seemingly less obvious causes limit the message of the paper?

presentation:

+ No doubt: this is a novel approach and an interesting piece of work. But the issues in question are also addressed in very many other papers. I think the list of references is too selective, seems almost random to me. Better relation and embedding into the existing literature, discussion of competing and related approaches will definitely improve the paper.

+ The observation (in the intro) "Our study confirms that during a normal period the market is very modular and heterogeneous, whereas during an instability (crisis) the market is more homogeneous, highly connected and less modular." is indeed very well known. It is thus a bit surprising that the next sentence starts with "These new insights ...". The new insights should be named here.

+ Sec 2 is way too tough on those with little background in advanced geometry. It so happens that I could understand it, but only because I once was interested in general relativity and the related differential geometry. The authors should be aware that the vast majority of the readers (particularly most practitioners) will not have enough background knowledge. Simple examples, figures etc may greatly help the dissemination of the methods and the results.

Decision letter (RSOS-201734.R0)

This year has been very difficult for everyone, and we want to take the opportunity to thank you for your continued support in 2020.

The Royal Society Open Science editorial office will be closed from the evening of Friday 18 December 2020 until Monday 4 January 2021. We will not be responding during this time. If you have received a deadline within this time period, please contact us as soon as possible to allow us to extend the deadline. If you receive any automated messages during this time asking you to meet a deadline, we offer apologies and invite you to respond after the festive period or during normal working hours.

With our best for a peaceful festive period and New Year, and we look forward to working with you in 2021.

Dear Dr Samal

The Editors assigned to your paper RSOS-201734 "Network geometry and market instability" have now received comments from reviewers and would like you to revise the paper in

accordance with the reviewer comments and any comments from the Editors. Please note this decision does not guarantee eventual acceptance.

Please submit your revised manuscript and required files (see below) no later than 21 days from today's (ie 22-Dec-2020) date. Note: the ScholarOne system will 'lock' if submission of the revision is attempted 21 or more days after the deadline. If you do not think you will be able to meet this deadline please contact the editorial office immediately.

on behalf of Dr Robert MacKay (Associate Editor) and Mark Chaplain (Subject Editor)
openscience@royalsociety.org

Associate Editor Comments to Author (Dr Robert MacKay):

Associate Editor: 1

Comments to the Author:

The reviewers have widely disparate views on this submission. I recommend a major revision to address the issues they raise.

Reviewer comments to Author:

Reviewer: 1

Comments to the Author(s)

It is known that geometrical measures and space-curvature are associated with network properties and the relation between correlation networks MSTs and other information filtering networks and the space curvature has been investigated previously [see Complex networks on hyperbolic surfaces Physica A 2005 Volume 346 20-26].

The fact that Ricci curvature is related with other network measures is not a surprise neither a novelty.

The question the authors should have addressed is whether it is useful to map simpler network topological properties into more complicated geometrical properties.

This paper fails to prove that there are insights in the Ricci curvature properties that cannot be uncovered with traditional network topological properties. Furthermore, the interpretation in terms of space-curvature does not provide any simpler or more intuitive perspective and therefore it results in a useless exercise.

Reviewer: 2

Comments to the Author(s)

This is a very interesting study. The authors put forward and apply advanced geometrical methods involving different curvatures to characterize financial markets. As these are nonstationary with an often abrupt dynamics, development of new methods can be a rewarding enterprise. In the present case, I am sure that the guiding ideas are an important novel input facilitating the analysis of data from various complex systems, including financial markets.

The manuscript certainly contains enough results to warrant publication in Royal Society Open Science.

There are a few points which the authors should address before publication:

research:

+ Minimum spanning trees are notoriously known for lack of robustness. Put differently: if I change the data a little bit (playing here, shaking there) I often get very different MST. Has this been taken into account? What is the impact on the results presented? Are they sensitive to the particular details of the MST? I do not find the present discussion sufficient. The authors showed that the "results are not very sensitive to the choice of overlapping or non-overlapping windows". Good! But this is not the same point.

+ In figs 3 and 4 the different measures show several peaks. In some cases they are related to drastic events in the markets, as indicated by dashed lines. Much more should be said on the other peaks. Are they spurious? If so, why? To which extent does the presence of these peaks with seemingly less obvious causes limit the message of the paper?

presentation:

+ No doubt: this is a novel approach and an interesting piece of work. But the issues in question are also addressed in very many other papers. I think the list of references is too selective, seems almost random to me. Better relation and embedding into the existing literature, discussion of competing and related approaches will definitely improve the paper.

+ The observation (in the intro) "Our study confirms that during a normal period the market is very modular and heterogeneous, whereas during an instability (crisis) the market is more homogeneous, highly connected and less modular." is indeed very well known. It is thus a bit surprising that the next sentence starts with "These new insights ..." The new insights should be named here.

+ Sec 2 is way too tough on those with little background in advanced geometry. It so happens that I could understand it, but only because I once was interested in general relativity and the

related differential geometry. The authors should be aware that the vast majority of the readers (particularly most practitioners) will not have enough background knowledge. Simple examples, figures etc may greatly help the dissemination of the methods and the results.

===PREPARING YOUR MANUSCRIPT===

===PREPARING YOUR REVISION IN SCHOLARONE===

- 1) One version identifying all the changes that have been made (for instance, in coloured highlight, in bold text, or tracked changes);
 - 2) A 'clean' version of the new manuscript that incorporates the changes made, but does not highlight them.
 - An individual file of each figure (EPS or print-quality PDF preferred [either format should be produced directly from original creation package], or original software format).
 - An editable file of each table (.doc, .docx, .xls, .xlsx, or .csv).
 - An editable file of all figure and table captions.
- Note: you may upload the figure, table, and caption files in a single Zip folder.
- Any electronic supplementary material (ESM).
 - If you are requesting a discretionary waiver for the article processing charge, the waiver form must be included at this step.
 - If you are providing image files for potential cover images, please upload these at this step, and inform the editorial office you have done so. You must hold the copyright to any image provided.
 - A copy of your point-by-point response to referees and Editors. This will expedite the preparation of your proof.

- Ensure that your data access statement meets the requirements at <https://royalsociety.org/journals/authors/author-guidelines/#data>. You should ensure that you cite the dataset in your reference list. If you have deposited data etc in the Dryad repository, please include both the 'For publication' link and 'For review' link at this stage.
- If you are requesting an article processing charge waiver, you must select the relevant waiver option (if requesting a discretionary waiver, the form should have been uploaded at Step 3 'File upload' above).
- If you have uploaded ESM files, please ensure you follow the guidance at <https://royalsociety.org/journals/authors/author-guidelines/#supplementary-material> to include a suitable title and informative caption. An example of appropriate titling and captioning may be found at https://figshare.com/articles/Table_S2_from_Is_there_a_trade-off_between_peak_performance_and_performance_breadth_across_temperatures_for_aerobic_sc_ope_in_teleost_fishes_/3843624.

Author's Response to Decision Letter for (RSOS-201734.R0)

See Appendix A.

Decision letter (RSOS-201734.R1)

Dear Dr Samal

On behalf of the Editors, we are pleased to inform you that your Manuscript RSOS-201734.R1 "Network geometry and market instability" has been accepted for publication in Royal Society Open Science subject to minor revision in accordance with the referees' reports. Please find the referees' comments along with any feedback from the Editors below my signature.

Please submit your revised manuscript and required files (see below) no later than 7 days from today's (ie 26-Jan-2021) date. Note: the ScholarOne system will 'lock' if submission of the revision is attempted 7 or more days after the deadline. If you do not think you will be able to meet this deadline please contact the editorial office immediately.

Best regards,

on behalf of Dr Robert MacKay (Associate Editor) and Mark Chaplain (Subject Editor)
openscience@royalsociety.org

Associate Editor Comments to Author (Dr Robert MacKay):

Thank you for your resubmission. I am satisfied with your responses to the reviewers and with your revised manuscript. The addition of figures in the ESM to illustrate the notions of Ricci curvature is also useful. I feel the paper makes a valuable contribution to the literature; time and experience will identify to what extent it is useful. I do have one question that I should have posed at the previous stage, I'm sorry: correlation networks are popular but is it not better to use "partial correlation", based on the inverse of the covariance matrix ("precision matrix")? The issue is that two nodes may be highly correlated but not so strongly if you condition on the states of the remaining nodes. The partial correlations detect direct connection as opposed to potentially indirect connection. If you would be willing to add a comment on this, I think it would be useful.

===PREPARING YOUR MANUSCRIPT===

===PREPARING YOUR REVISION IN SCHOLARONE===

Author's Response to Decision Letter for (RSOS-201734.R1)

See Appendix B.

Decision letter (RSOS-201734.R2)

Dear Dr Samal,

It is a pleasure to accept your manuscript entitled "Network geometry and market instability" in its current form for publication in Royal Society Open Science. The comments of the reviewers who reviewed your manuscript are included at the foot of this letter.

Best regards,

on behalf of Dr Robert MacKay (Associate Editor) and Mark Chaplain (Subject Editor)
openscience@royalsociety.org

Associate Editor Comments to Author (Dr Robert MacKay):

Thank you for responding to my supplementary question. I am happy with your response and the added paragraph to the paper. I recommend acceptance.

Appendix A

Dated: 21st January 2021

To,
The Editor,
Royal Society Open Science

Dear Prof. MacKay,

We would like to resubmit our revised manuscript for your favourable consideration towards publication in the *Royal Society Open Science*. The details of the manuscript are as follows:

- Manuscript ID: RSOS-201734
- Title of the paper: Network geometry and market instability
- Authors: Areejit Samal, Hirdesh K. Pharasi, Sarath Jyotsna Ramaia, Harish Kannan, Emil Saucan, Jürgen Jost, and Anirban Chakraborti

We are grateful to you for giving us an opportunity to revise and resubmit the manuscript, and to the referees for their critical comments and useful suggestions, which we have duly taken into account in the revised version.

Reviewer 1 had been critical about the novelty as well as the insights from discrete Ricci curvatures as opposed to those obtained from application of traditional network topological measures. We have tried to clarify all the issues raised by the referee to the best of our ability and added corresponding text as well as a few references in the revised version to justify our response. We strongly believe that our interdisciplinary approach is very important for the dissemination of knowledge and adds significant advancement as well as new insights to this developing field.

Reviewer 2 had raised an important point about the robustness and stability of the network construction method employed here. We have addressed it and shown that the results are not sensitive to small amounts of noise or random reshuffling of data. Regarding the second point, we have clarified and explained that the ‘other peaks’ are neither spurious nor are they false positives. These are some events that are not well understood or recorded in the financial literature (or listed in internet sources); and we are glad that our methodology is picking up such events in accordance with some other papers that we have referred to in the revised version. The other points about presentation have been taken care of while revising the introduction thoroughly, where we have also included several additional references. Finally, to improve the description of the discrete Ricci curvatures, we have added four schematic figures with simple examples illustrating the computation of the Ricci curvatures in graphs in the electronic supplementary material of the revised manuscript.

We reiterate that we have taken into account all the suggestions given by referees, and revised the manuscript accordingly. We have also corrected a few typographical and grammatical errors during this revision. *We are uploading the revised manuscript in two versions, one with changes highlighted in Red and other with no changes highlighted. In addition, we enclose a point-wise response to comments by reviewers and modifications made in the revised manuscript in response to comments by referees.*

Kindly note that the data used for this study was downloaded from the public domain of Yahoo Finance (<https://finance.yahoo.com/>). All relevant data and codes for this study have been uploaded and made publicly available via a GitHub repository: <https://github.com/asamallab/StockMarkNetIndicator>.

We also certify that none of the material has been published or is under consideration for publication elsewhere.

Looking forward to receiving your favourable decision soon!

Thanking you,

Sincerely yours,

Dr. Areejit Samal

The Institute of Mathematical Sciences (IMSc),
Chennai 600113, India

Email: asamal@imsc.res.in

&

Prof. (Dr.) Anirban Chakraborti

School of Computational and Integrative Sciences,
Jawaharlal Nehru University, New Delhi-110067, India

E-mail: anirban@jnu.ac.in

(For the authors)

Point-wise response to comments by reviewers on the manuscript

Network geometry and market instability

Manuscript ID: RSOS-201734

We provide below a point-wise response to each comment made by the two reviewers and list the modifications made in the revised manuscript to address the comments. Reviewer comments are reproduced in **brown** while our responses are in **blue**.

We thank the reviewers for the constructive comments which have helped improve the revised manuscript. We hope that the editor and the reviewers will be satisfied with these changes and recommend the revised manuscript for publication in Royal Society Open Science.

Associate Editor Comments to Author (Dr Robert MacKay):

Associate Editor: 1

Comments to the Author:

The reviewers have widely disparate views on this submission. I recommend a major revision to address the issues they raise.

Answer:

We sincerely thank the reviewers for their critical comments and the Editor for giving us the opportunity to revise and resubmit our manuscript. We have made major revisions in the manuscript, as per the suggestions, and responded to ALL the criticisms below.

Reviewer comments to Author:

Reviewer: 1

Comments to the Author(s)

It is known that geometrical measures and space-curvature are associated with network properties and the relation between correlation networks MSTs and other information filtering networks and the space curvature has been investigated previously [see Complex networks on hyperbolic surfaces Physica A 2005 Volume 346 20-26].

The fact that Ricci curvature is related with other network measures is not a surprise neither a novelty.

The question the authors should have addressed is whether it is useful to map simpler network topological properties into more complicated geometrical properties.

This paper fails to prove that there are insights in the Ricci curvature properties that cannot be uncovered with traditional network topological properties. Furthermore, the interpretation in terms of space-curvature does not provide any simpler or more intuitive perspective and therefore it results in a useless exercise.

Answer:

Firstly, we would like to address the following comment by the referee: “The fact that Ricci curvature is related with other network measures is not a surprise neither a novelty. The question the authors should have addressed is whether it is useful to map simpler network topological properties into more complicated geometrical properties.”

In this manuscript, we have *applied* four notions of discrete Ricci curvature, namely, Ollivier-Ricci, Forman-Ricci, Haantjes-Ricci and Menger-Ricci, to correlation-based financial networks. These four notions of discrete Ricci curvatures have been introduced to the domain of complex networks in previous publications in reputed peer-reviewed journals.

We would like to reiterate that the present manuscript is only the first study to apply three of these four discrete Ricci curvatures, namely, Forman-Ricci, Haantjes-Ricci and Menger-Ricci, to financial networks.

To elaborate, Ollivier's seminal work [Ollivier *Comptes Rendus Mathématique* 345: 643–646 (2007)] has led to a discretization of Ricci curvature, namely, Ollivier-Ricci curvature, which has since received significant attention, and Ollivier-Ricci curvature has been used extensively to obtain theoretical results on graphs and application to real-world networks. Among theoretical results obtained by mathematicians via study of Ollivier-Ricci curvature in graphs, we would like to highlight for example the work of Shing-Tung Yau [e.g., Lin, Yau. *Math. Res. Lett.* 17: 343–356 (2010)] and one of the authors of this manuscript, Jürgen Jost [e.g., Bauer et al. *Math. Res. Lett.* 19: 1185–1205 (2012)]. Among applications of Ollivier-Ricci curvature to real-world networks, we would like to highlight the work of Allen Tannebaum on cancer networks [Sandhu et al. *Scientific Reports* 5: 12323 (2015)], brain networks [Farooq et al. *Nature communications* 10: 4937 (2019)] and financial networks [Sandhu et al. *Science Advances* 2: e1501495 (2016)], and the work of Jie Gao on communication networks [Ni et al. *INFOCOM* (2015)] and community detection [Ni et al. *Scientific Reports* 9: 9984 (2019)] in networks. In short, we are not the first ones to apply Ollivier-Ricci curvature to networks nor have we claimed any novelty in this respect in this manuscript.

On the other hand, some of us have successfully ported three other notions of discrete Ricci curvatures, namely, Forman-Ricci [Sreejith et al. *JSTAT* P063206 (2016)], Haantjes-Ricci and Menger-Ricci [Saucan et al. *Network Science* (2020) DOI:10.1017/nws.2020.42], to the domain of networks. Our previous publications on these discrete Ricci curvatures have analyzed extensively the relationship between these edge-centric measures and other traditional network measures. Further, our work on Forman-Ricci curvature has received significant attention since publication in 2016 while our work on Haantjes-Ricci and Menger-Ricci curvature appeared only a couple of months ago.

Importantly, as highlighted in section 2 of this manuscript, the four discrete notions of Ricci curvature capture different aspects of the classical notion of Ricci curvature, and therefore, they are likely to capture different aspects of the network structure. Specifically, Ollivier-Ricci curvature captures the volume growth property of the classical notion while Forman-Ricci curvature captures the geodesic dispersal property of the classical notion, and thus, theoretically the two notions of discrete Ricci curvatures have no particular reason to give same insights in a specific network. Surprisingly, an empirical analysis however by some of us finds a high correlation between the two discretizations of Ricci curvature in real-world networks [Samal et al. *Scientific Reports* 8: 8650 (2018)].

Further, Forman-Ricci curvature is a local computation while Ollivier-Ricci curvature is a global computation, and thus, there are several computational advantages in employing Forman-Ricci curvature to complex networks. This is an additional motivation for this study which extends the work on Ollivier-Ricci curvature by Allen Tannenbaum in stock market networks [Sandhu et al. *Science Advances* 2: e1501495 (2016)] to multiple notions of discrete Ricci curvature and two different stock markets.

In summary, it is rather unfortunate that we are being forced to defend the novelty of discrete Ricci curvatures (including Ollivier-Ricci and Forman-Ricci) which have been successfully applied to diverse real-world networks in several previous

publications in reputed peer-reviewed journals (mostly, by other scientists). While this manuscript is not concerned about the relationship between discrete Ricci curvatures and standard network measures but rather application to financial networks, we think that the offhand dismissal by the reviewer towards any such significant and/or novel observations in previous work is definitely not deserved.

Secondly, we would like to address the following comment by the referee: “It is known that geometrical measures and space-curvature are associated with network properties and the relation between correlation networks MSTs and other information filtering networks and the space curvature has been investigated previously [see Complex networks on hyperbolic surfaces Physica A 2005 Volume 346 20-26].”

In response to the above comments by the referee, we have added the following paragraph in the Introduction section of the revised manuscript:

“It is noteworthy that in the present paper, the term ‘curvature’ refers to four notions of discrete Ricci curvature investigated here, which are as such intrinsic curvatures, and not extrinsic curvatures as has been considered elsewhere in the context of complex networks (see e.g., Aste et al. [54]). Recall that extrinsic geometry is given by embedding the networks in a suitable ambient space (which in practice is the hyperbolic plane or space), and thereafter, the geometric properties induced by the embedding space are studied (see, e.g. [55]). While this approach is intuitive and conducive to simple illustrations, such network embeddings are distorting, except for the special case of isometric embeddings. In contrast, the intrinsic approach to networks is independent of any specific embedding, and hence, of the necessary additional computations and any distortion. Moreover, such an intrinsic approach allows for the independent study of such powerful tools as the Ricci flow, without the vagaries associated with the embedding in an ambient space of certain dimension (see, e.g. [56]). Furthermore, the Ollivier-Ricci curvature has been employed to show that the ‘backbone’ of certain real-world networks is indeed tree-like, hence intrinsically hyperbolic [49]. Specific to financial networks, Sandhu et al. [50] have shown that Ollivier-Ricci curvature, which is of course an intrinsic curvature, presents a powerful tool in the detection of financial market crashes. In this work, we have considered three additional notions [43,55] of discrete Ricci curvature for the study of financial networks.”

As highlighted in the above paragraph in the revised manuscript, the reviewer has missed the point that we have applied intrinsic curvatures rather than extrinsic curvatures to financial networks. Therefore, we have included the above paragraph in the revised manuscript to stress the advantages of employing intrinsic curvatures to networks. In addition, we would like to highlight that the discrete Ricci curvatures employed in this manuscript are edge-centric rather than node-centric. For example, certain notions of Gauss curvature for networks are node-centric rather than edge-centric.

In summary, it is rather unfortunate that the referee has not recognized the difference between intrinsic and extrinsic curvatures for networks.

Thirdly, we would like to address the following comment by the referee: “This paper fails to prove that there are insights in the Ricci curvature properties that cannot be uncovered with traditional network topological properties. Furthermore, the interpretation in terms of space-curvature does not provide any simpler or more intuitive perspective and therefore it results in a useless exercise.”

We are disappointed that the referee has completely overlooked the new aspects and motivation for this study. In this context, we would like to reproduce the following paragraph in the Results section of the manuscript:

“A major goal of this research is to evaluate different notions of discrete Ricci curvature for their ability to unravel the structure of complex financial networks and serve as indicators of market instabilities. Previously, Sandhu et al. [50] have analyzed the USA S&P-500 market over a period of 15 years (1998-2013) to show that the average Ollivier-Ricci (OR) curvature of edges (ORE) in threshold networks increases during periods of financial crisis. Here, we extend the analysis by Sandhu et al. [50] to (a) two different stock markets, namely, USA S&P-500 and Japanese Nikkei-225, (b) a span of 32 years (1985-2016), (c) four traditional market indicators (namely, index log-returns r , mean market correlation μ , volatility of the market index r estimated using $GARCH(1,1)$ process, and risk σ_P corresponding to the minimum risk Markowitz portfolio of all the stocks in the market), and (d) four notions of discrete Ricci curvature for networks. Since discretizations of Ricci curvature are unable to capture the entire properties of the classical Ricci curvature defined on continuous spaces, the four discrete Ricci curvatures evaluated here can shed light on different properties of analyzed networks [48]. In particular, some of us have introduced another discretization, Forman-Ricci (FR) curvature, to the domain of networks [43]. Note that OR curvature captures the volume growth property of classical Ricci curvature while FR curvature captures the geodesic dispersal property [48]. Nevertheless, our empirical analysis has shown that the two discrete notions, OR and FR curvature, are highly correlated in model and real-world networks [48]. Importantly, in large networks, computation of the OR curvature is intensive while that of the FR curvature is simple as the later depends only on immediate neighbours of an edge [48]. Therefore, we started by investigating the ability of FR curvature to capture the structure of complex financial networks.”

As highlighted clearly in the above paragraph in our manuscript, Sandhu et al. [Science Advances 2: e1501495 (2016)] have previously applied Ollivier-Ricci curvature to study correlation-based networks for USA S&P-500 market, and thereafter, have shown that Ollivier-Ricci curvature is an excellent indicator of market fragility. Given the publication by Sandhu et al. (in a reputed journal), we believe our work is an worthwhile enterprise due to following reasons as highlighted in the above paragraph of the manuscript:

- 1) One, we have extended the application of discrete Ricci curvatures to two different stock markets, namely, USA S&P-500 and Japanese Nikkei-225.
- 2) Two, we consider a span of 32 years (1985-2016) instead of 15 years by Sandhu et al.
- 3) Three, we have compared our results with four traditional market indicators namely, index log-returns r , mean market correlation μ , volatility of the market index r estimated using $GARCH(1,1)$ process, and risk σ_P corresponding to the minimum risk Markowitz portfolio of all the stocks in the market. Further, we have also compared the results from discrete Ricci curvatures with several simple network measures such as average degree, clustering coefficient, average path length, etc. which have not been considered in Sandhu et al.
- 4) Four, we have considered four discrete Ricci curvatures for networks instead of only Ollivier-Ricci curvature in Sandhu et al.

In short, based on the previous work of Sandhu et al. in a reputed journal, it was natural to explore other notions of discrete Ricci curvature in financial networks.

Further, our result that Forman-Ricci curvature has a slightly higher correlation with traditional market indicators than Ollivier-Ricci curvature studied by Sandhu et al. in both stock markets, justifies such an exploration in financial networks. As Forman-Ricci is significantly easier to compute than Ollivier-Ricci curvature, we believe this manuscript reports an important result for the practitioners.

We hope that our detailed response has convinced the referee that her/his characterization of this work as ‘useless exercise’ is also a comment on the previous work by Sandhu et al. and all publications on discrete Ricci curvatures for graphs/networks to date. While this is unfortunate, we believe it is important to both disagree with the referee and clearly highlight the previous contributions by several researchers in the broad area of network geometry.

Finally, we are very surprised and to some extent disappointed that the reviewer did not make a single comment about the significance of our results and applicability of our methods in the context of financial markets, which is a substantial part and the focus of our manuscript. We have been very open and transparent about our findings and presented all the results in a way that will assist any researcher, novice or expert, to compare the properties of discrete Ricci curvatures as well as the traditional network measures in the context of financial markets, especially during bubbles and crashes. For example, in contrast to Sandhu et al., we have contrasted the results from application of discrete Ricci curvatures to stock market networks with those obtained from simple network measures such as number of edges, average degree, clustering coefficient etc. In the Results section of the manuscript, we have clearly reported this contrast as follows: *“From Figures 3 and 4, it is seen that NE and CE are also excellent indicators of market instabilities. In fact, we find that common network measures such as number of edges, edge density, average degree, average shortest path length, graph diameter, average clustering coefficient and modularity are also good indicators of market instabilities (ESM Figure S8).”* It is possible that the referee may view this transparent reporting of marginal gain via application of discrete Ricci curvatures in contrast to simple network measures, as a lack of novelty in this manuscript. However, we strongly believe that this transparency is very important for the dissemination of knowledge in this type of interdisciplinary research and it is a significant advancement in this developing field. Note that Sandhu et al. had not presented such a detailed contrast of Ollivier-Ricci curvature vis-a-vis several simple network measures.

Reviewer: 2

Comments to the Author(s)

This is a very interesting study. The authors put forward and apply advanced geometrical methods involving different curvatures to characterize financial markets. As these are nonstationary with an often abrupt dynamics, development of new methods can be a rewarding enterprise. In the present case, I am sure that the guiding ideas are an important novel input facilitating the analysis of data from various complex systems, including financial markets.

The manuscript certainly contains enough results to warrant publication in Royal Society Open Science.

Answer:

We are very grateful to the reviewer for finding the study interesting and important, and also providing an extremely positive and constructive feedback.

There are a few points which the authors should address before publication:

research:

+ Minimum spanning trees are notoriously known for lack of robustness. Put differently: if I change the data a little bit (playing here, shaking there) I often get very different MST. Has this been taken into account? What is the impact on the results presented? Are they sensitive to the particular details of the MST? I do not find the present discussion sufficient. The authors showed that the "results are not very sensitive to the choice of overlapping or non-overlapping windows". Good! But this is not the same point.

Answer:

We thank the reviewer for raising this very important issue of robustness and stability. We do understand that a MST can be sensitive to outliers or reshuffling of data in the correlation matrix. However, it has been shown in earlier papers [Micciche et al. *Physica A* 324: 66-73 (2003), Onnela et al. *Physica A* 324: 247-252 (2003)] that the topological properties of the MSTs can be quite robust and stable.

To test the robustness of results in the current paper, we have added small amounts of Gaussian noise to the empirical correlation matrices, and reproduced the results with the topological properties as well as the geometric curvature measures. We have shown that the results are not sensitive to small amounts of noise or random reshuffling of data. Also, as we stated before, the results were not very sensitive to the small changes of window size (exclusion and inclusion of small amounts of data).

We have added a few lines on this robustness analysis in the conclusions section of the revised manuscript. Further, the results from this robustness analysis are shown in a new figure (Electronic Supplementary Material Figure S12) in the revised manuscript.

+ In figs 3 and 4 the different measures show several peaks. In some cases they are related to drastic events in the markets, as indicated by dashed lines. Much more should be said on the other peaks. Are they spurious? If so, why? To which extent does the presence of these peaks with seemingly less obvious causes limit the message of the paper?

Answer:

Once again, we acknowledge the reviewer for raising an important point. The "other peaks" are neither spurious nor are they false positives. They do represent significant market events, and we are glad that our methodology is picking up such events. However, unlike the major crashes and bubbles which are well-documented in the financial literature (or listed in internet sources), many of these interesting events are not well understood or recorded in the literature. In fact the financial markets are often driven by endogenous and exogenous factors, and some of the authors of the present paper have tried to study and characterize such market events (exogenous shocks, bubble bursts, anomalies, etc.) in a series of papers [Pharasi et al. *New Journal of Physics* 20: 103041 (2018), Chakraborti et al. *New Journal of Physics* 22: 063043. (2020), Chakraborti et al. *Journal of Physics: Complexity* 2: 015002 (2020)]. We have clarified this and added a few lines in the conclusions section of the revised manuscript.

presentation:

+ No doubt: this is a novel approach and an interesting piece of work. But the issues in question are also addressed in very many other papers. I think the list of references is too selective, seems almost random to me. Better relation and embedding into the existing literature, discussion of competing and related approaches will definitely improve the paper.

Answer:

We apologize that we were a bit selective about the references and related approaches. We have added several references on related approaches and explained them in the thoroughly revised introduction in the revised manuscript. However, please note there may be a few important papers which we may have unintentionally missed.

+ The observation (in the intro) "Our study confirms that during a normal period the market is very modular and heterogeneous, whereas during an instability (crisis) the market is more homogeneous, highly connected and less modular." is indeed very well known. It is thus a bit surprising that the next sentence starts with "These new insights ..." The new insights should be named here.

Answer:

We have benefited from this feedback and thoroughly revised the introduction in revised manuscript accordingly.

+ Sec 2 is way too tough on those with little background in advanced geometry. It so happens that I could understand it, but only because I once was interested in general relativity and the related differential geometry. The authors should be aware that the vast majority of the readers (particularly most practitioners) will not have enough background knowledge. Simple examples, figures etc may greatly help the dissemination of the methods and the results.

Answer:

We are grateful for this feedback. Indeed, it might be a bit difficult for a broad audience to understand advanced concepts in differential geometry including Ricci curvatures. As per the suggestion, we have added 4 schematic diagrams in the Electronic Supplementary Material (Figure S1, S2, S3 and S4) of the revised manuscript that may assist in the dissemination of the methods and the results. We hope that these supplementary figures along with the schematic illustration in the main text will help the reader to grasp the necessary background on discrete Ricci curvatures for future applications.

Finally, we would like to point out that for the benefit of readers and researchers, all relevant data and codes for this study, including those used to compute the four discrete Ricci curvatures, have been uploaded and made publicly available via the GitHub repository:

<https://github.com/asamallab/StockMarkNetIndicator>.

We hope that the reviewers and the editor will be satisfied with the modifications made in the revised manuscript and recommend publication in the current form.

Appendix B

Dated: 26th January 2021

To,
The Editor,
Royal Society Open Science

Dear Prof. MacKay,

We thank you very much for the provisional acceptance of our manuscript “Network geometry and market instability” (Manuscript ID: RSOS-201734) in the *Royal Society Open Science*, and your valuable question. We would like to resubmit our revised manuscript after addition of suitable comments in the text along with relevant references in this revised version.

We append at the end of this letter your comment (reproduced in brown) and our response along with details of modification made during this revision in blue. *We are uploading the revised manuscript in two versions, one with changes highlighted in Red and other with no changes highlighted.*

Sincerely yours,

Dr. Areejit Samal

The Institute of Mathematical Sciences (IMSc), Chennai 600113, India

Email: asamal@imsc.res.in

&

Prof. (Dr.) Anirban Chakraborti

School of Computational and Integrative Sciences, Jawaharlal Nehru University, New Delhi-110067, India

E-mail: anirban@jnu.ac.in

(For the authors)

Point-wise response to comments by editor

Comment by Editor:

Thank you for your resubmission. I am satisfied with your responses to the reviewers and with your revised manuscript. The addition of figures in the ESM to illustrate the notions of Ricci curvature is also useful. I feel the paper makes a valuable contribution to the literature; time and experience will identify to what extent it is useful. I do have one question that I should have posed at the previous stage, I'm sorry: correlation networks are popular but is it not better to use "partial correlation", based on the inverse of the covariance matrix ("precision matrix")? The issue is that two nodes may be highly correlated but not so strongly if you condition on the states of the remaining nodes. The partial correlations detect direct connection as opposed to potentially indirect connection. If you would be willing to add a comment on this, I think it would be useful.

Response:

We thank you for raising this point. We agree with you that partial correlations detect direct as opposed to potentially indirect connection. In the Econophysics literature [Kenett et al. (2010); San Miguel et al. EPJ Special Topics (2012); Millington & Niranjana, Applied Network Science (2020)], and in few of our earlier papers [Sharma et al. (2017); Chakraborti, J. Phys. Complexity (2020)], researchers have used partial correlations for analyzing the

dynamics and constructing networks of stock markets. Partial correlations are particularly relevant when people study eigenvalue spectra (market, group and random modes) or centrality measures, by first filtering out the spurious correlations. However, it has been observed [Millington & Niranjana, *Applied Network Science* (2020); Chakraborti et al. *J. Phys. Complexity* (2020)] that partial correlations are less successful in picking the cluster or group dynamics, and the networks arising from partial correlations are also less stable. In this paper, we are more interested in the market indicators and the use of discrete curvatures as generic indicators, for which we prefer to work with the correlation matrices. We have added this point along with relevant references in the conclusion section of our revised manuscript.